# Scaling limit of the staggered six-vertex model with $U_q\big(\mathfrak{sl}(2)\big)$ invariant boundary conditions

**Holger Frahm[⋆], Sascha Gehrmann[†] and Gleb A. Kotousov[‡]**

Institut für Theoretische Physik, Leibniz Universität Hannover
Appelstraße 2, 30167 Hannover, Germany

⋆ frahm@itp.uni-hannover.de ,   † sascha.gehrmann@itp.uni-hannover.de ,
‡ gleb.kotousov@itp.uni-hannover.de

## Abstract

We study the scaling limit of a statistical system, which is a special case of the integrable inhomogeneous six-vertex model. It possesses $U_q\big(\mathfrak{sl}(2)\big)$ invariance due to the choice of open boundary conditions imposed. An interesting feature of the lattice theory is that the spectrum of scaling dimensions contains a continuous component. By applying the ODE/IQFT correspondence and the method of the Baxter $Q$ operator the corresponding density of states is obtained. In addition, the partition function appearing in the scaling limit of the lattice model is computed, which may be of interest for the study of non-rational CFTs in the presence of boundaries. As a side result of the research, a simple formula for the matrix elements of the $Q$ operator for the general, integrable, inhomogeneous six-vertex model was discovered, that has not yet appeared in the literature. It is valid for a certain one parameter family of diagonal open boundary conditions in the sector with the $z$-projection of the total spin operator being equal to zero.

## 1  Introduction

Onsager's solution of the square lattice Ising model [1] opened an era in the application of integrable systems to the study of phase transitions. Exactly solvable lattice models in 2D have been found to exhibit an interesting array of effects and powerful analytic/numerical techniques are available to explore them. Among such phenomena are, for instance, exactly marginal deformations [2–4], strong/weak coupling dualities [5], and the appearance of extended conformal symmetry in the scaling limit [6], which have collectively inspired and refined our understanding of (conformal) QFT. Of special mention is the Pott's model, a generalization of the 2D Ising model. In the antiferromagnetic case it possesses a phase, where the ground state degeneracy is macroscopic with its logarithm being proportional to the system size; similar to fractons that have recently attracted attention (see, e.g., [7] for a review). This observation was originally made in the 80's by Berker and Kadanoff [8]. The universal properties were further studied in [9] via the mapping of the critical Potts to the six-vertex model [10] (see also [11]) and the application of the methods of Yang-Baxter integrability and 2D CFT.

In the work [8] the Potts model is taken to be homogeneous and isotropic, so that the coupling between any two nearest neighbour pairs of 'spins' is the same. One can consider an anisotropic version defined on the square lattice, which has two coupling constants — one associated to the vertical and the other to the horizontal edges of the lattice that join the neighbouring spins together. As explained in ref. [12], focusing on the curve in the parameter space where the model is critical leads one to the so-called staggered six-vertex model, which is a special case of the integrable, inhomogeneous six-vertex model introduced by Baxter in [13]. It gets its name from the fact that the inhomogeneities are distributed along the square lattice in a checkerboard (staggered) pattern. In addition, in the case of the antiferromagnetic Potts model, they are fixed to a special value for which the system is 'self-dual', i.e., possesses an extra $\mathcal{Z}_2$ symmetry.

The critical behaviour of the staggered six-vertex model at the self-dual point was considered in the works [12,14]. Valuable results about the spectrum of scaling dimensions were obtained by studying the low energy spectrum of the Hamiltonian, which is expressed in terms of a logarithmic derivative of the two row transfer-matrix. The Hamiltonian, unlike the transfer-matrix, is given by a sum of operators, which act locally on $\mathscr{V}_{2L} = \mathbb{C}_1^2 \otimes \mathbb{C}_2^2 \otimes \ldots \otimes \mathbb{C}_{2L}^2$, where $2L$ is the number of lattice columns. The precise formula reads as $\mathbb{H} = -\frac{\mathrm{i}}{q^2 - q^{-2}} \sum_{J=1}^{2L} \mathcal{O}_J$ with[1]

$$
\begin{aligned}
\mathcal{O}_J \; = \; & \left(q - q^{-1}\right)^2 \sigma_J^z \sigma_{J+1}^z + 2\left(\sigma_J^x \sigma_{J+2}^x + \sigma_J^y \sigma_{J+2}^y + \sigma_J^z \sigma_{J+2}^z\right) \\
& - \left(q - q^{-1}\right)\left(\sigma_J^z\left(\sigma_{J+1}^x \sigma_{J+2}^x + \sigma_{J+1}^y \sigma_{J+2}^y\right) - \left(\sigma_J^x \sigma_{J+1}^x + \sigma_J^y \sigma_{J+1}^y\right)\sigma_{J+2}^z\right) - \left(q^2 + q^{-2}\right)\hat{\mathbf{1}} .
\end{aligned}
$$
(1.1)

---

[1]The formula for the Hamiltonian (7.6) in the work [15] is identical to the one given above except that the overall sign in front of the term $\propto (q - q^{-1})$ in the second line of eq. (1.1), containing the product of three Pauli matrices, is flipped. This comes about because that paper uses the different convention for the quantum space: $\mathscr{V}_{2L} = \mathbb{C}_{2L}^2 \otimes \mathbb{C}_{2L-1}^2 \otimes \ldots \otimes \mathbb{C}_1^2$, see eq. (2.1) therein. The two Hamiltonians are related via the similarity transformation $\mathcal{U} : \mathcal{U}^2 = 1$, which acts on the local spin operators as $\mathcal{U} \sigma_J^A \mathcal{U} = \sigma_{2L-J+1}^A$.

Here $\sigma_J^A$ with $A = x, y, z$ are the Pauli matrices that act on site $J$ subject to periodic boundary conditions $\sigma_{J+2L}^A = \sigma_J^A$, while the parameter $q$ is known as the anisotropy. The system turns out to be critical when $q$ is a unimodular number and different universal behaviour occurs depending on whether $\arg(q) \in (0, \frac{\pi}{2})$ or $\arg(q) \in (\frac{\pi}{2}, \pi)$. The regime

$$|q| = 1 \qquad \text{and} \qquad \arg(q) \in (0, \tfrac{\pi}{2}) \tag{1.2}$$

has attracted the most amount of attention. The reason for this is that the corresponding spectrum of scaling dimensions was found to possess a continuous component [14].

The subsequent study of the regime (1.2) saw a remarkable interchange of ideas between statistical mechanics and formal high energy theory. On the one hand, the conjecture from ref. [16], that the scaling limit of the lattice system is governed by the 2D Euclidean black hole sigma model introduced in refs. [17–19], uses the results of [20] and its development [21] which come from the string theory literature. On the other, the detailed study of the vertex model performed in [15] led to the solution of the spectral problem for the 2D Euclidean black hole CFT, including the computation of the density of states of the continuous spectrum. Perhaps the most surprising output of the research is the following. While it has been confirmed that one half of the partition function arising in the scaling limit of the vertex model with (quasi-)periodic boundary conditions coincides with the partition function of the 2D Euclidean black hole sigma model on the torus, the original conjecture of [16] has been refined. It was proposed in [15] that a part of the Hilbert space of the lattice model in the scaling limit should coincide with the pseudo-Hilbert space of the black hole sigma model with Lorentzian signature.

The above mentioned works all focus on the case when the lattice is (quasi)-periodic in the horizontal direction. In the recent papers [22, 23], motivated by the possibility of making precise contact with $D$-brane constructions of non-compact boundary CFTs [24, 25], the statistical system has been considered with certain integrable, open boundary conditions imposed. In this case the Hamiltonian is given by

$$\mathbb{H} = -\frac{\mathrm{i}}{q^2 - q^{-2}} \Big( \sum_{J=1}^{2L-2} \mathcal{O}_J - (q + q^{-1})\big(\sigma_1^x \sigma_2^x + \sigma_1^y \sigma_2^y + \sigma_{2L-1}^x \sigma_{2L}^x + \sigma_{2L-1}^y \sigma_{2L}^y\big)$$

$$-(q^2 - q^{-2})(\sigma_{2L}^z - \sigma_1^z) - 2(\sigma_1^z \sigma_2^z - \hat{\mathbf{1}}) + (q^2 + q^{-2})(\sigma_{2L-1}^z \sigma_{2L}^z - \hat{\mathbf{1}}) \Big), \tag{1.3}$$

where $\mathcal{O}_J$ is defined in eq. (1.1). A special feature of such a choice of boundary terms is that the model possesses $U_q\big(\mathfrak{sl}(2)\big)$ symmetry. To explain, notice that $\mathbb{H}$ commutes with the $z$-projection of the total spin operator:

$$\mathbb{S}^z = \frac{1}{2} \sum_{J=1}^{2L} \sigma_J^z. \tag{1.4}$$

One may check that it also commutes with

$$\mathbb{S}_q^{\pm} = (\mp \mathrm{i}) \sum_{J=1}^{2L} \Big( \prod_{\ell=J+1}^{2L} q^{-\frac{\sigma_\ell^z}{2}} \Big)(-1)^J \, \sigma_J^{\pm} \Big( \prod_{\ell=1}^{J-1} q^{+\frac{\sigma_\ell^z}{2}} \Big), \tag{1.5}$$

which, together with $\mathbb{S}^z$, satisfy the defining relations of the $U_q\big(\mathfrak{sl}(2)\big)$ algebra:[2]

$$\big[\mathbb{S}^z, \mathbb{S}_q^{\pm}\big] = \pm \mathbb{S}_q^{\pm}, \qquad \big[\mathbb{S}_q^+, \mathbb{S}_q^-\big] = \frac{q^{2\mathbb{S}^z} - q^{-2\mathbb{S}^z}}{q - q^{-1}}. \tag{1.6}$$

---

[2]The factor $(\mp\mathrm{i})$ in eq. (1.5) together with the term $(-1)^J$ appearing in the summand may be removed via a similarity transformation by a diagonal matrix.

As a result, the eigenstates of the Hamiltonian form irreps of this algebra. These are labelled by the eigenvalues of the Casimir,

$$2\mathbb{C} = (q + q^{-1})[\mathbb{S}^z]_q^2 + \mathbb{S}_q^+ \mathbb{S}_q^- + \mathbb{S}_q^- \mathbb{S}_q^+ \,, \tag{1.7}$$

which are given by $[\mathcal{S}]_q[\mathcal{S}+1]_q$ with integer $\mathcal{S} = 0, 1, 2, \ldots, L$ (here we use the standard notation $[m]_q = (q^m - q^{-m})/(q - q^{-1})$). The presence of $U_q\big(\mathfrak{sl}(2)\big)$ symmetry simplifies the diagonalization problem for $\mathbb{H}$. In particular, one can restrict to the $(2L)!/(L!)^2$ dimensional subspace spanned by the eigenstates of the operator $\mathbb{S}^z$, whose eigenvalues are zero.

It was observed in refs. [22,23] that the spectrum of scaling dimensions of the staggered six-vertex model with $U_q\big(\mathfrak{sl}(2)\big)$ invariant open boundary conditions possesses a continuous component in the regime (1.2). However, the corresponding density of states had not been obtained. This was among the open problems that inspired our research.

In this paper we perform a systematic study of the low energy spectrum of the Hamiltonian (1.3) at large system size $L \gg 1$. It is carried out via a mixture of methods, including a numerical analysis of the Bethe Ansatz equations as well as the powerful analytical technique of the ODE/IQFT correspondence. It turns out that the ODEs describing the scaling limit fall within the class of differential equations considered in refs. [15,26], where the universal behaviour of the vertex model with quasi-periodic boundary conditions imposed was studied. They are described in detail in sec. 3.2 below. As for the Bethe Ansatz equations, they read as

$$\left(\frac{1 + \zeta_m^2 q^{+2}}{1 + \zeta_m^2 q^{-2}}\right)^{2L} = q^{4+4\mathcal{S}} \prod_{\substack{j=1 \\ j \neq m}}^{L-\mathcal{S}} \frac{\big(\zeta_j - q^{+2}\zeta_m\big)\big(1 - q^{+2}\zeta_m\zeta_j\big)}{\big(\zeta_j - q^{-2}\zeta_m\big)\big(1 - q^{-2}\zeta_m\zeta_j\big)} \,, \tag{1.8}$$

where $\mathcal{S}$ stands for the $U_q\big(\mathfrak{sl}(2)\big)$ total spin of the state. Having at hand a solution set $\{\zeta_m\}_{m=1}^{L-\mathcal{S}}$ to the above algebraic system, the energy of the corresponding state is computed via the formula:

$$\mathcal{E} = \sum_{m=1}^{L-\mathcal{S}} \frac{4\mathrm{i}(q^2 - q^{-2})}{\zeta_m^2 + \zeta_m^{-2} + q^2 + q^{-2}} \,. \tag{1.9}$$

Notice that eqs. (1.8) are invariant upon making the transformation $\zeta_m \mapsto \zeta_m^{-1}$ of any one of the Bethe roots. This allows one, without loss of generality, to assume that

$$|\zeta_m| \le 1 \qquad (m = 1, 2, \ldots, L - \mathcal{S}). \tag{1.10}$$

The paper is organized as follows. In section 2 we present a formula for the matrix elements of the Baxter $Q$ operator that was used in our numerical work. It is valid in the sector $S^z = 0$ and for a one parameter family of open boundary conditions. The outcomes of our study of the low energy spectrum of the Hamiltonian (1.3) in the regime (1.2) is given in section 3. The first subsection thereof, for the most part, is a review of the results contained in refs. [22,23]. In the next two, the ODE/IQFT correspondence is described and, on the basis of this, the so-called 'quantization condition' is obtained. The latter is what allows us to perform a full characterization of the low energy space of states of the lattice model, which is detailed in section 4. In section 5, the formula for the partition function appearing in the scaling limit of the lattice model is given, which may be of interest to those studying non-rational boundary CFT. The last section is devoted to a discussion and includes a summary of the main results.

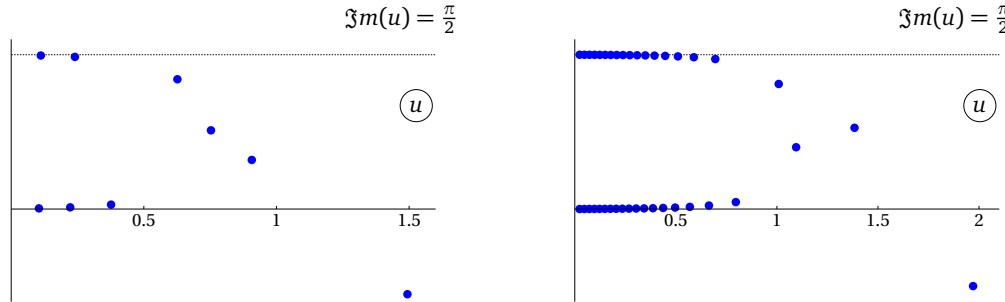

Figure 1: Displayed is the pattern of Bethe roots in the complex $u = -\frac{1}{2}\log(\zeta)$ plane for a low energy Bethe state that was generically chosen. For the left panel $L = 10$ and the set $\{\zeta_m\}$ was obtained from the eigenvalue of the $Q$ operator computed on the state. The latter was used to seed an RG trajectory $\{|\Psi_L\rangle\}$ and, on the right panel, shown are the Bethe roots for the corresponding state with $L = 40$. The trajectory is labelled by $\mathcal{S} = 1$, while the anisotropy parameter was taken to be $q = e^{\frac{10\pi i}{49}}$.

## 2 The $Q$ operator

In analysing the low energy spectrum of the Hamiltonian of a critical lattice system in the scaling limit one meets with immediate issues. An important one concerns the construction of RG trajectories, where low energy stationary states at different lattice sizes $L$ are grouped into families $\{|\Psi_L\rangle\}$. It is clear how to define the RG flow for the ground state or, for that matter, the lowest energy states in the disjoint sectors of the Hilbert space, say, the sector with given eigenvalue of $\mathbb{S}^z$ (1.4). However, assigning an $L$ dependence to a low energy excited state and then continuing $|\Psi_L\rangle$ to $L \gg 1$ seems to be a non-trivial task.

In the case at hand the model is integrable and the construction of RG trajectories is facilitated by the Bethe Ansatz solution. A formulation of the procedure is provided in ref. [26], see also the work [15]. The low energy stationary states can be chosen to be the Bethe states, which are simultaneous eigenvectors of the full family of operators commuting with the Hamiltonian. They are labeled by the Bethe roots $\{\zeta_m\}$, which solve the Bethe Ansatz equations (1.8). The eigenvalues of the commuting operators for the Bethe state are given in terms of the corresponding $\{\zeta_m\}$, as in formula (1.9) for the energy, and their computation does not require any explicit diagonalization. Suppose one has at hand the Bethe roots for a low energy Bethe state for a lattice of $L = L_{\text{in}}$ sites. The state $|\Psi_{L_{\text{in}}+2}\rangle$ is specified such that the pattern of Bethe roots qualitatively remains the same. These can be obtained by numerically solving the Bethe Ansatz equations (1.8), where the initial approximation for the iterative solution is constructed from the Bethe roots corresponding to $|\Psi_{L_{\text{in}}}\rangle$. By iterating this procedure an RG trajectory $\{|\Psi_L\rangle\}$ for increasing $L$ is obtained.

A method is required in order to extract the Bethe roots corresponding to the state $|\Psi_{L_{\text{in}}}\rangle$ which seeds the RG trajectory. Typically for $|\Psi_{L_{\text{in}}}\rangle$ generic, $\{\zeta_m\}$ are complex numbers that do not resemble a simple pattern in the complex plane, see Fig. 1 for an example. Finding all of the possible solution sets of the Bethe Ansatz equations (1.8) even for $L_{\text{in}} \lesssim 10$ is impossible to carry out on a modern laptop because the system is too complicated. Moreover, searching for the set $\{\zeta_m\}$ by applying the Newton method to (1.8), trying out various initial approximations, is time consuming and not guaranteed to work.

In our study of the spin chain, to find the Bethe roots corresponding to a generic low energy state $|\Psi_{L_{\text{in}}}\rangle$ systematically, we used the technique based on the notion of the Baxter $Q$ operator $\mathbb{Q}(\zeta)$ [27]. The latter commutes with itself for different values of the spectral parameter as

well as the transfer-matrix,

$$\mathbb{Q}(\zeta): \qquad [\mathbb{Q}(\zeta), \mathbb{Q}(\zeta')] = [\mathbb{Q}(\zeta), \mathbb{T}(\zeta')] = 0. \qquad (2.1)$$

In addition, its eigenvalues for a Bethe state is a polynomial in $\zeta$, whose zeroes coincide with the corresponding solution set to the Bethe Ansatz equations (see, e.g., formula (2.18) below). For a small number of lattice sites it is possible to obtain the first few hundred low lying stationary states of the spin chain via an explicit diagonalization of the Hamiltonian. For each of them, the corresponding set $\{\zeta_m\}$ is extracted by computing the eigenvalue of $\mathbb{Q}(\zeta)$ (some further details can be found in section 18 of ref. [15]). In turn, the RG trajectory $\{|\Psi_L\rangle\}$ is continued from $L = L_{\text{in}}$ to large $L$ via the procedure outlined above.

Below we present the explicit formula for the $Q$ operator that was used in our analysis. It allowed us to go beyond the results of the previous papers [22, 23]. Our formula was obtained based on the results of ref. [28], which is part of an interesting group of recent papers [28–31] appearing in the mathematics literature. Among other things, they give $\mathbb{Q}(\zeta)$ as a trace of a monodromy matrix over a $q$-oscillator representation for models associated with the rational [29] and trigonometric [30, 31] $R$-matrix for $\mathfrak{sl}(2)$ with a two parameter family of open Boundary Conditions (BCs). The matrix elements of $\mathbb{Q}(\zeta)$ from [28] take the form of an infinite sum, which converges only in a certain parameteric domain that excludes the model with $U_q\big(\mathfrak{sl}(2)\big)$ invariant BCs we are considering. As such, some analysis was required in order to bring the expression to a form, which is literally applicable to the case at hand and is efficient for numerical purposes.

Let's consider the more general case of a lattice system with an arbitrary set of inhomogeneities $\{\eta_J\}_{J=1}^{2L}$. Moreover, for technical reasons, we'll take a one parameter family of open BCs depending on $\epsilon$, such that the $U_q\big(\mathfrak{sl}(2)\big)$ invariant case is recovered at

$$\epsilon = 0 \qquad\qquad \Big(U_q\big(\mathfrak{sl}(2)\big) \text{ invariant BCs}\Big). \qquad (2.2)$$

Rather than following the conventions of ref. [23], we'll use the multiplicative spectral parameter $\zeta$ and arrange the definitions such that the matrix elements of both $\mathbb{T}(\zeta)$ and $\mathbb{Q}(\zeta)$ are mainifestly polynomials in $\zeta$. The $R$-matrix reads as

$$R(\zeta) = \begin{pmatrix} q - q^{-1}\zeta & 0 & 0 & 0 \\ 0 & 1-\zeta & q-q^{-1} & 0 \\ 0 & (q-q^{-1})\zeta & 1-\zeta & 0 \\ 0 & 0 & 0 & q-q^{-1}\zeta \end{pmatrix}. \qquad (2.3)$$

It can be interpreted as an operator $R(\zeta) = R_{I,J}(\zeta)$ acting on the tensor product $\mathbb{C}_I^2 \otimes \mathbb{C}_J^2$, where $\mathbb{C}_J^2$ with $J = 0$ stands for the auxiliary space, while for $J = 1, \ldots, 2L$ it is the $J$-th factor of the quantum space

$$\mathscr{V}_{2L} = \mathbb{C}_1^2 \otimes \mathbb{C}_2^2 \otimes \ldots \otimes \mathbb{C}_{2L}^2. \qquad (2.4)$$

The transfer-matrix, which is graphically depicted in Fig. 2, is given by

$$\mathbb{T}(\zeta q^{-1}) = q^{-2L} \operatorname{tr}_0\Big(K_0^+(\zeta) R_{0,2L}(\zeta\,\eta_{2L}^{-1}) \ldots R_{0,1}(\zeta\,\eta_1^{-1}) K_0^-(\zeta) R_{1,0}(\zeta\,\eta_1) \ldots R_{2L,0}(\zeta\,\eta_{2L})\Big). \qquad (2.5)$$

Here the trace is taken over the auxiliary space, while $K^\pm$ stand for the diagonal matrices

$$K^-(\zeta) = \begin{pmatrix} 1+\zeta\epsilon & 0 \\ 0 & \zeta^2 + \zeta\epsilon \end{pmatrix}, \qquad K^+(\zeta) = \begin{pmatrix} q^{-2}\zeta + \epsilon & 0 \\ 0 & \zeta^{-1} + q^{-2}\epsilon \end{pmatrix}. \qquad (2.6)$$

Figure 2: A graphical representation of the transfer-matrix for the inhomogeneous six-vertex model. Open BCs are imposed, as indicated by the presence of the reflection matrices $K^{\pm}$ given by (2.6) acting in the two-dimensional auxiliary space.

It is straightforward to check that $\mathbb{T}(\zeta)$ is a polynomial of order $4L+2$ in the spectral parameter and satisfies the conditions

$$\mathbb{T}(0) = \epsilon\left(q^{2\mathbb{S}^z} + q^{-2\mathbb{S}^z}\right), \qquad\qquad \mathbb{T}(\zeta^{-1}) = \zeta^{-4L-2}\,\mathbb{T}(\zeta)\,. \qquad (2.7)$$

To get back the case of the staggered six-vertex model with $U_q\big(\mathfrak{sl}(2)\big)$ invariant BCs, at the self-dual point, one should fix the inhomogeneities $\eta_J$ and the extra parameter $\epsilon$ to be

$$\eta_J = \mathrm{i}\,(-1)^J \qquad\quad (J = 1, 2, \ldots 2L), \qquad\qquad \epsilon = 0\,. \qquad (2.8)$$

We will use the notation[3]

$$\mathbb{T}^{(0)}(\zeta) \equiv \mathbb{T}(\zeta)\big|_{\epsilon=0}\,. \qquad (2.9)$$

For our purposes it is sufficient to focus on the sector where the eigenvalue of the $z$-projection of the total spin operator is zero, i.e., $S^z = 0$. This is because for the model possessing $U_q\big(\mathfrak{sl}(2)\big)$ invariance, the states come in multiplets $\mathcal{M}_\mathcal{S}$ each of which has a representative in that sector. Let the tuples $(a_1 a_2 \ldots a_{2L})$ and $(b_1 b_2 \ldots b_{2L})$ with $a_J, b_J = \pm$ be the input/output indices for the space $\mathscr{V}_{2L}$ (2.4). To present the formula for the $Q$ operator, introduce

$$\begin{pmatrix} \left[A(\zeta;m)\right]_+^+ & \left[A(\zeta;m)\right]_+^- \\ \left[A(\zeta;m)\right]_-^+ & \left[A(\zeta;m)\right]_-^- \end{pmatrix} = \begin{pmatrix} q^m & q^m \\ \zeta q^{-m+1} & q^{-m} \end{pmatrix}, \qquad (2.10)$$

$$\begin{pmatrix} \left[\widetilde{A}(\zeta;m)\right]_+^+ & \left[\widetilde{A}(\zeta;m)\right]_+^- \\ \left[\widetilde{A}(\zeta;m)\right]_-^+ & \left[\widetilde{A}(\zeta;m)\right]_-^- \end{pmatrix} = \begin{pmatrix} q^m & \zeta q^{m+2} \\ q^{-m-1} & q^{-m} \end{pmatrix}. \qquad (2.11)$$

---

[3]The transfer-matrix as defined in formula (2.5) for arbitrary inhomogenieties $\eta_J$ and $\epsilon = 0$ is related to $\tau(u)$ given by (2.9) of ref. [23] via a similarity transformation by a diagonal matrix and an overall multiplicative factor. Namely,

$$\mathbb{T}^{(0)}(\zeta) = 2^{4L}\mathrm{e}^{-4Lu-2u}\bigg(\prod_{J=1}^{2L}\rho(-u + \tfrac{\mathrm{i}\gamma}{2} - \delta_J)\bigg)\,\mathcal{U}\,\tau(u)\,\mathcal{U}^{-1}\,.$$

Here $\mathcal{U} = G_1(\delta_1)\otimes\cdots\otimes G_{2L}(\delta_{2L})$ with $G(u) = \mathrm{diag}(1, \mathrm{e}^{-u})$, while the parameters $u, \gamma, \delta_J$ need to be identified with $\zeta, q, \eta_J$ as

$$\zeta = \mathrm{e}^{-2u}, \qquad q = \mathrm{e}^{\mathrm{i}\gamma}, \qquad \eta_J = \mathrm{e}^{-2\delta_J}\,.$$

Also, the function $\rho(u)$ is given by $\rho(u) = \tfrac{1}{2}\big(\cos(2\gamma) - \cosh(2u)\big)$.

The matrix elements of $\mathbb{Q}(\zeta)$, valid in the sector $S^z = 0$, are given by[4]

$$\left[\mathbb{Q}(\zeta)\right]_{a_1\ldots a_{2L}}^{b_1\ldots b_{2L}} = \sum_{c_1\ldots c_{2L}=\pm} q^{(S_c^z)^2}(\epsilon q^2\zeta)^{S_c^z}\prod_{J=1}^{2L}\left[A(\zeta\eta_J^{-1},m_{b,J})\right]_{c_J}^{b_J}\left[\widetilde{A}(\zeta\eta_J,m_{a,J+1})\right]_{a_J}^{c_J}. \quad (2.12)$$

Here the symbol $S_c^z$, which should not be confused with the eigenvalue of the $z$-projection of the total spin operator, stands for

$$S_c^z = \frac{1}{2}\sum_{J=1}^{2L}c_J \quad (2.13)$$

and provides a natural grading for the sum over $c_J$. The internal indices $\{m_{a,J}, m_{b,J}\}$ come from the product over the auxiliary space. They are fixed completely by the ice-rule to be

$$m_{x,J} = \frac{1}{2}\sum_{\ell=J}^{2L}(x_\ell - c_\ell) \qquad (J=1,\ldots,2L+1;\ x=a,b). \quad (2.14)$$

Despite the presence of the factor $\zeta^{S_c^z}$ in formula (2.12), where the exponent can be negative, the matrix elements of $\mathbb{Q}(\zeta)$ turn out to be polynomials in $\zeta$ of degree $2L$. Moreover, it is straightforward to show that they satisfy:

$$\mathbb{Q}(\zeta^{-1}) = \zeta^{-2L}\,\mathbb{Q}(\zeta), \qquad \mathbb{Q}(0) = \mathbf{1}. \quad (2.15)$$

The following comment is in order regarding the relation of the $Q$ operator whose matrix elements are given by (2.12) with that studied in ref. [28]. In fact, in the latter work two $Q$ operators $\mathbb{Q}^{(a)}$ with $a = 1, 2$ are introduced. In order to specialize to the one parameter family of open boundary conditions being considered here one should restrict the parameters $\bar{\epsilon}_\pm$ and $\epsilon_\pm$ in that paper as $\epsilon_+/\epsilon_- = \bar{\epsilon}_+/\bar{\epsilon}_- = \epsilon$. Then, choosing a representation for the $q$ oscillator algebra, formula (5.13) from [28] provides an expression for the matrix elements of the $Q$ operators in terms of an infinite sum $\sum_{m\geq 0}g(m)$. For the sector $S^z = 0$ and generic values of the parameters this sum diverges for both $\mathbb{Q}^{(1)}$ and $\mathbb{Q}^{(2)}$ so that eq. (5.13) becomes inapplicable. This is because the summand $g(m)$ tends to a finite, nonvanishing limit as $m \to \infty$. Our formula (2.12) essentially coincides with $\lim_{m\to\infty}g(m)$. The limiting value is the same whether or not we started from $\mathbb{Q}^{(1)}$ or $\mathbb{Q}^{(2)}$ so one may only obtain a single $Q$ operator in this way. Remarkably, we have checked for small lattice sizes $L = 2, 3, 4, \ldots$ that $\mathbb{Q}(\zeta)$ from (2.12) obeys the commutativity conditions (2.1) as well as the $TQ$ relation with the transfer-matrix

$$\left(1-\zeta^2\right)\mathbb{Q}(\zeta)\mathbb{T}(\zeta) = \left(\epsilon + q^{+1}\zeta\right)\left(1+\zeta q^{+1}\epsilon\right)f(q^{-1}\zeta)\,\mathbb{Q}\left(\zeta q^{+2}\right) \quad (2.16)$$

$$+ \left(\epsilon + q^{-1}\zeta\right)\left(1+\zeta q^{-1}\epsilon\right)f(q^{+1}\zeta)\,\mathbb{Q}\left(\zeta q^{-2}\right) \qquad (S^z = 0).$$

Here

$$f(\zeta) = (1-\zeta^2)\prod_{J=1}^{2L}\left(\zeta - \eta_J^{-1}\right)\left(\zeta - \eta_J\right), \quad (2.17)$$

---

[4]Written in index notation, the transfer matrix (2.5) takes the form:

$$\left[\mathbb{T}(\zeta q^{-1})\right]_{a_1\ldots a_{2L}}^{b_1\ldots b_{2L}} = q^{-2L}\sum_{c_1\ldots c_{2L}=\pm}\sum_{\substack{\alpha_1\ldots\alpha_{2L+1}=\pm\\\beta_1\ldots\beta_{2L+1}=\pm}}\left[K^+(\zeta)\right]_{\alpha_{2L+1}}^{\beta_{2L+1}}\left[R(\zeta\eta_{2L}^{-1})\right]_{\alpha_{2L}c_{2L}}^{\alpha_{2L+1}b_{2L}}\ldots\left[R(\zeta\eta_1^{-1})\right]_{\alpha_1 c_1}^{\alpha_2 b_1}$$

$$\left[K^-(\zeta)\right]_{\beta_1}^{\alpha_1}\left[R(\zeta\eta_1)\right]_{a_1\beta_2}^{c_1\beta_1}\ldots\left[R(\zeta\eta_{2L})\right]_{a_{2L}\beta_{2L+1}}^{c_{2L}\beta_{2L}}.$$

Here $\left[K^\pm(\zeta)\right]_\alpha^\beta$ with $\alpha,\beta = \pm$ stand for the entries of the diagonal matrices (2.6) and similar for $\left[R(\zeta)\right]_{aa}^{\beta b}$, e.g., $\left[R(\zeta)\right]_{-+}^{+-} = q - q^{-1}$ and $\left[R(\zeta)\right]_{+-}^{-+} = (q-q^{-1})\zeta$.

while the values of the inhomogeneities $\eta_J$ and parameter $\epsilon$ are assumed to be generic. The Bethe Ansatz equations for the integrable model follow as usual: One considers both sides of eq. (2.16) evaluated on a common eigenvector. In view of (2.15), the eigenvalues of $\mathbb{Q}(\zeta)$ take the form

$$Q(\zeta) = \prod_{j=1}^{L} (1 - \zeta/\zeta_j)(1 - \zeta\zeta_j). \tag{2.18}$$

Combining the above with (2.16) and setting $\zeta = \zeta_m$ into that formula, one arrives at the coupled system of algebraic equations for the Bethe roots:

$$\prod_{J=1}^{2L} \frac{\left(\zeta_m q^{+1} - \eta_J^{-1}\right)\left(\zeta_m q^{+1} - \eta_J\right)}{\left(\zeta_m q^{-1} - \eta_J^{-1}\right)\left(\zeta_m q^{-1} - \eta_J\right)} = \frac{\left(\epsilon + q^{+1}\zeta_m\right)\left(q^{+1}\epsilon + \zeta_m^{-1}\right)}{\left(\epsilon + q^{-1}\zeta_m\right)\left(q^{-1}\epsilon + \zeta_m^{-1}\right)} \tag{2.19}$$

$$\times \quad q^2 \prod_{\substack{j=1 \\ j\neq m}}^{L} \frac{\left(\zeta_j - q^{+2}\zeta_m\right)\left(1 - q^{+2}\zeta_m\zeta_j\right)}{\left(\zeta_j - q^{-2}\zeta_m\right)\left(1 - q^{-2}\zeta_m\zeta_j\right)} \qquad (S^z = 0).$$

These are equivalent to the Bethe Ansatz equations obtained in the original work [32], specialized to the sector $S^z = 0$ and with $\xi^+ = -\xi^-$. Also, the parameters should be identified as $(q, \eta_J, \epsilon) \mapsto (\mathrm{e}^\eta, \mathrm{e}^{-2u_J}, \mathrm{e}^{2\xi^\pm})$, while $\zeta_m \mapsto \mathrm{e}^{-2v_m}$. We find it surprising that a formula like (2.12) exists and its further exploration may be worthwhile. At the same time, since our work is focused on the study of the scaling limit of an integrable lattice system, we believe that it is not the place to do this here.

Some care is needed in taking the limit $\epsilon \to 0$ of the $Q$ operator. It is clear from the explicit formula (2.12) that the matrix elements of $\mathbb{Q}(\zeta)$ generically diverge due to the presence of the factor $\epsilon^{S_c^z}$, where the exponent may be negative. This is a manifestation of the $U_q\bigl(\mathfrak{sl}(2)\bigr)$ invariance possessed by the model at the point $\epsilon = 0$, so that states in different sectors of $S^z$ form multiplets of the symmetry group that have the same eigenvalue of $\mathbb{Q}(\zeta)$. In ref. [33] a similar phenomenon was studied in the context of the XXX spin chain with twisted boundary conditions controlled by the parameter $\phi$. At $\phi = 0$ the model possesses global $\mathfrak{su}(2)$ symmetry and the matrix elements of the $Q$ operator become infinite. It was explained in that paper how to take the limit $\phi \to 0$ of $\mathbb{Q}(\zeta)$ so that one obtains a well defined result. The discussion is readily adapted to the case at hand.

Recall that the quadratic Casimir for the $U_q\bigl(\mathfrak{sl}(2)\bigr)$ algebra is given by:

$$2\mathbb{C} = (q + q^{-1})[\mathbb{S}^z]_q^2 + \mathbb{S}_q^+ \mathbb{S}_q^- + \mathbb{S}_q^- \mathbb{S}_q^+ , \tag{2.20}$$

where $\mathbb{S}^z$ is defined in formula (1.4) in the introduction, while for arbitrary values of the inhomogeneities,

$$\mathbb{S}_q^\pm = \sum_{J=1}^{2L} \left( \prod_{\ell=J+1}^{2L} q^{-\frac{\sigma_\ell^z}{2}} \right) \eta_J^{\mp 1} \sigma_J^\pm \left( \prod_{\ell=1}^{J-1} q^{+\frac{\sigma_\ell^z}{2}} \right). \tag{2.21}$$

The eigenvalues of $\mathbb{C}$ are given by

$$[\mathcal{S}]_q [\mathcal{S} + 1]_q \tag{2.22}$$

with $\mathcal{S} = 0, 1, 2, \dots, L$ . One can consider $\mathcal{S}$ as an operator, which for an eigenstate of the quadratic Casimir with eigenvalue $[\mathcal{S}]_q [\mathcal{S} + 1]_q$ gives back the non-negative integer $\mathcal{S} \geq 0$. Then, following the work [33], it turns out that the limit

$$\mathbb{Q}^{(0)}(\zeta) = \lim_{\epsilon \to 0} \epsilon^{\frac{\mathcal{S}}{2}} \mathbb{Q}(\zeta; \epsilon) \epsilon^{\frac{\mathcal{S}}{2}} \tag{2.23}$$

exists and yields the $Q$ operator for the inhomogeneous six-vertex model with $U_q\big(\mathfrak{sl}(2)\big)$ invariant BCs in the sector $S^z = 0$. The commutativity condition (2.1) and the $TQ$ relation (2.16) with the substitutions $\big(\mathbb{Q}, \mathbb{T}\big) \mapsto \big(\mathbb{Q}^{(0)}, \mathbb{T}^{(0)}\big)$ are satisfied provided, for the latter formula, one sets $\epsilon = 0$. We note, however, that the normalisation (2.15) no longer holds true. Instead,

$$\mathbb{Q}^{(0)}(\zeta)|\Psi_L\rangle = C\,\zeta^{\mathcal{S}} \prod_{j=1}^{L-\mathcal{S}} (1 - \zeta/\zeta_j)(1 - \zeta\zeta_j)|\Psi_L\rangle\,, \tag{2.24}$$

where $C$ is a constant depending only on $q$ and $\mathcal{S}$.

The $TQ$ relation specialized to $\epsilon = 0$, combined with (2.24), leads to the Bethe Ansatz equations

$$\prod_{J=1}^{2L} \frac{\big(\zeta_m q^{+1} - \eta_J^{-1}\big)\big(\zeta_m q^{+1} - \eta_J\big)}{\big(\zeta_m q^{-1} - \eta_J^{-1}\big)\big(\zeta_m q^{-1} - \eta_J\big)} = q^{4+4\mathcal{S}} \prod_{\substack{j=1 \\ j\neq m}}^{L-\mathcal{S}} \frac{\big(\zeta_j - q^{+2}\zeta_m\big)\big(1 - q^{+2}\zeta_m\zeta_j\big)}{\big(\zeta_j - q^{-2}\zeta_m\big)\big(1 - q^{-2}\zeta_m\zeta_j\big)}\,. \tag{2.25}$$

Upon taking the inhomogeneities to be as in (2.8) one gets back eq. (1.8) that appeared in the introduction. Notice that (2.25) also follows from (2.19) by taking $\epsilon \to 0$ and assuming that $\mathcal{S}$ of the roots $\zeta_j$ with $j \neq m$ vanish in this limit.

# 3 Low energy spectrum in the scaling limit

## 3.1 Preliminaries

The low energy spectrum of the Hamiltonian of a 1D critical quantum spin chain at large system size contains important information about the universal behaviour of the model [35–38]. As such, upon the construction of the RG trajectories $\{|\Psi_L\rangle\}$, the corresponding eigenvalue of the Hamiltonian, $E(L)$, for $L \gg 1$ is one of the first quantities that may be studied. For the staggered six-vertex model with $U_q\big(\mathfrak{sl}(2)\big)$ invariant boundary conditions in the critical regime $q = \mathrm{e}^{\mathrm{i}\gamma}$ with $\gamma \in (0, \frac{\pi}{2})$ the energy of a certain class of low energy states was analyzed in the works [22, 23]. The leading and sub-leading behaviour is described as $E \asymp L e_\infty + f_\infty + o(1)$, where the specific bulk energy $e_\infty$ and surface contribution to the energy $f_\infty$ were obtained in ref. [23] within the root density approach:

$$e_\infty = -\int_{-\infty}^{+\infty} \mathrm{d}\omega\, \frac{\sinh(\frac{\pi\omega}{2n+4})}{\sinh(\frac{\pi\omega}{4})\cosh(\frac{n\pi\omega}{4n+8})} \tag{3.1}$$

$$f_\infty = 4\tan(\tfrac{\pi}{n+2}) + \int_{-\infty}^{\infty} \mathrm{d}\omega\, \frac{\cosh(\frac{\pi\omega}{4n+8})\sinh(\frac{\pi\omega(n-1)}{4n+8})}{\sinh(\frac{\pi\omega}{4})\cosh(\frac{n\pi\omega}{4n+8})}\,.$$

Here and below the anisotropy $q$ is swapped for the parameter $n > 0$ according to the relation

$$q = \mathrm{e}^{\mathrm{i}\gamma}: \qquad \gamma = \frac{\pi}{n+2} \qquad (n > 0)\,. \tag{3.2}$$

The next order term in the asymptotic expansion of the energy goes as $1/L$. Unlike $e_\infty$ and $f_\infty$, which are the same for all the low energy states, the coefficient for the $1/L$ correction depends on the particular RG trajectory $\{|\Psi_L\rangle\}$ one is considering. Among other things, it contains the conformal dimension characterising the scaling limit of that state. In the works [22, 23] the following asymptotic formula was proposed:

$$E \asymp L\,e_\infty + f_\infty + \frac{\pi v_{\mathrm{F}}}{L}\left(\frac{p^2}{n+2} + \frac{b^2}{n} - \frac{1}{12} + \mathtt{d}\right) + o(L^{-1-\varepsilon})\,. \tag{3.3}$$

The Fermi velocity $v_\mathrm{F}$ is non-universal and depends on the overall multiplicative normalization of the Hamiltonian. In our conventions for $\mathbb{H}$, one has

$$v_\mathrm{F} = 2 + \frac{4}{n} \ . \tag{3.4}$$

The term $p$ is shorthand for the expression

$$p = \frac{1}{2}\big(2\mathcal{S} + 1 + \mathtt{w}(n+2)\big) \qquad \text{with} \qquad \mathtt{w} = -1\ , \tag{3.5}$$

where $\mathcal{S}$ is the $U_q\big(\mathfrak{sl}(2)\big)$ spin of $|\Psi_L\rangle$, while the notation $\mathtt{d}$ stands for 'descendent' and is a non-negative integer $\mathtt{d} = 0, 1, 2, \dots$ . Also, the correction term $o(L^{-1-\varepsilon})$ contains an infinitesimally small $\varepsilon > 0$.

The quantity $b$ in the asymptotic formula (3.3) requires special comment. First of all, it turns out to depend on the system size $b = b(L)$. In ref. [23] it was found that $b(L)$ is related to the eigenvalue $B$ of the so-called 'quasi-shift' operator $\mathbb{B}$ computed on $|\Psi_L\rangle$. This operator commutes with the transfer-matrix and coincides, up to an overall factor,[5] with $\big(\mathbb{T}^{(0)}(iq^{-1})\big)^2$, see also ref. [39]. The precise relation reads as

$$b(L) = \frac{n}{2\pi} \log\big(\sqrt{B}\,\big) \ . \tag{3.6}$$

Here $\sqrt{B}$ is given in terms of the Bethe roots corresponding to a Bethe state as

$$\sqrt{B} = \prod_{m=1}^{L-\mathcal{S}} \frac{\big(\zeta_m + iq^{-1}\big)\big(\zeta_m - iq\big)}{\big(\zeta_m - iq^{-1}\big)\big(\zeta_m + iq\big)} \ . \tag{3.7}$$

The asymptotic formula (3.3) for the low energy spectrum should be understood with $b$, therein, substituted for $b(L)$ obtained via (3.6).

Some comment is required on the choice of the branch for the logarithm in the expression for $b(L)$ (3.6). For all the trajectories we constructed, it turned out that consistency with the asymptotic formula for the energy (3.3) requires that:

$$-\frac{n}{2} < \Im m\big(b(L)\big) \le \frac{n}{2} \ . \tag{3.8}$$

The question of which of the boundaries $\Im m\big(b(L)\big) = \pm\frac{n}{2}$ to include in the domain of $b(L)$ does not matter for the following reason. The only RG trajectories of the spin chain which were observed such that $\Im m\big(b(L)\big) \to \pm\frac{in}{2}$ as $L \to \infty$ had vanishing real part in the scaling limit. The typical pattern of Bethe roots for one of these is depicted in Fig. 3. In this case, one notes that the asymptotic formula for the energy (3.3) yields the same result for $\lim_{L\to\infty} L\big(E - Le_\infty - f_\infty\big)$ regardless of whether $\lim_{L\to\infty} b(L)$ coincides with $-\frac{in}{2}$ or $+\frac{in}{2}$.

The appearance of a term like $b = b(L)$ in the $1/L$ corrections to $E(L)$, that depends on the low energy state, was originally observed in the context of 1D quantum spin chains with periodic boundary conditions imposed [14, 40]. The problem of extracting the spectrum of conformal dimensions for these models was studied in detail in the later works [15, 16]. The analysis therein carries over to the staggered six-vertex model with open, $U_q\big(\mathfrak{sl}(2)\big)$ invariant boundary conditions we are considering. As an illustration, let's discuss it first for the class of states that were considered in the papers [22, 23].

---

[5]The factor is given by $-\frac{q^{4L+2}}{1+q^2} f^2(iq^{-2})$ with $f$ being the function defined in (2.17).

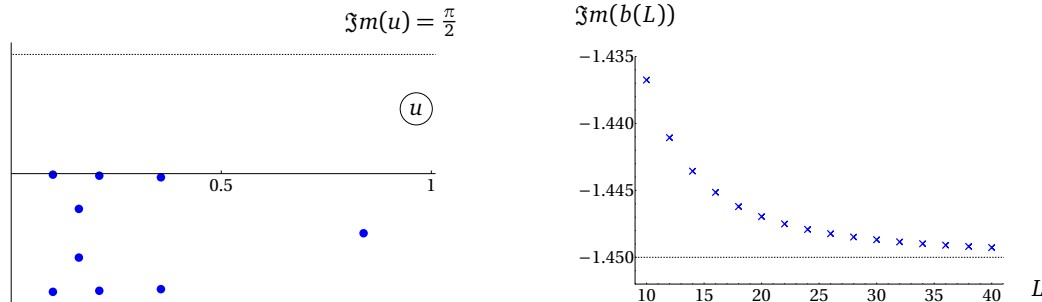

Figure 3: Displayed is numerical data for an RG trajectory where $b(L)$ tends to $-\frac{i n}{2}$. The left panel depicts the pattern of Bethe roots in the complex $u = -\frac{1}{2} \log(\zeta)$ plane for the state $|\Psi_L\rangle$ with $L = 10$. The right panel is a plot of $b(L)$ as a function of $L$, where the dashed line represents its limiting value. Here $\mathcal{S} = 1$ and $n$, which parameterizes $q$ as in (3.2), is given by $n = 2.9$.

There exist solutions to the Bethe Ansatz equations (1.8) such that all the Bethe roots are real. Different solution sets are distinguished by the integer $\mathfrak{m}$, which stands for the difference between the number of roots lying on the positive and negative real axes (see Fig.4):

$$\{\zeta_m\}_{m=1}^M = \{\zeta_m^{(+)}\}_{m=1}^{M+\frac{\mathfrak{m}}{2}} \cup \{\zeta_m^{(-)}\}_{m=1}^{M-\frac{\mathfrak{m}}{2}} \qquad \text{with} \qquad \zeta_m^{(\pm)} \gtrless 0 \qquad (M = L - \mathcal{S}). \tag{3.9}$$

Comparison of the energy computed from the solution set $\{\zeta_m\}$ via eq. (1.9) with that coming from the direct diagonalization of the Hamiltonian, for small lattice sizes, shows that the corresponding Bethe states are low energy states of the spin chain provided $\mathfrak{m} \ll L$. The large $L$ asymptotic of the energy turns out to obey (3.3) with $\mathsf{d} = 0$ so long as there are no significant gaps in the distribution of Bethe roots along the positive and negative rays $\zeta \lessgtr 0$. Computing the value of $b(L)$ from the definition (3.7), (3.6) one finds it to be a real number, which turns out to satisfy the large $L$ asymptotic behaviour [23]

$$b_{\mathfrak{m}}(L) = \frac{\pi \mathfrak{m}}{2 \log(L)} + O\big(1/(\log L)^2\big) \qquad (\mathfrak{m} - \text{fixed}). \tag{3.10}$$

This way as $L$ becomes large, $\Delta b_{\mathfrak{m}}(L) = b_{\mathfrak{m}+2}(L) - b_{\mathfrak{m}}(L) \propto 1/\log L$, so that the values of $b_{\mathfrak{m}}(L)$ are densely distributed in some segment of the real line. The latter is given by $(-b_{\mathfrak{m}_{\max}}, +b_{\mathfrak{m}_{\max}})$, where $\mathfrak{m}_{\max}(L) \ll L$ is the maximum value of the integer $\mathfrak{m}$ such that the state with Bethe roots (3.9) is still of low energy. Assuming that $\mathfrak{m}_{\max}$ grows faster than $\log(L)$ as $L \to \infty$, this segment becomes the entire real line in the scaling limit.

When assigning an $L$ dependence $|\Psi_L\rangle$ to the class of states discussed above, it is tempting to keep the integer $\mathfrak{m}$ fixed. Then, in view of formula (3.10), the value of $b(L)$ would go to zero as $L \to \infty$. However, there is another way of organizing the RG flow of the states. One may increase $\mathfrak{m}$ as $\sim \log(L)$ so that the value of $b(L)$ tends to a finite, non zero limit as $L \to \infty$. Such an RG trajectory would be characterized by

$$s = \operatorname*{slim}_{L \to \infty} b(L), \tag{3.11}$$

which can be arranged to be an arbitrary real number. Here and below we use the symbol slim for 'scaling limit' to emphasize that there is additional non-trivial input involved in taking the number of sites to infinity. Let's compare the asymptotic formula for the energy (3.3) for the RG trajectory with the general CFT prediction for a lattice system with open boundary conditions imposed [35, 36]:

$$E \asymp L e_\infty + f_\infty + \frac{\pi v_F}{L}\Big(\Delta - \frac{c}{24}\Big) + o(L^{-1-\varepsilon}), \tag{3.12}$$

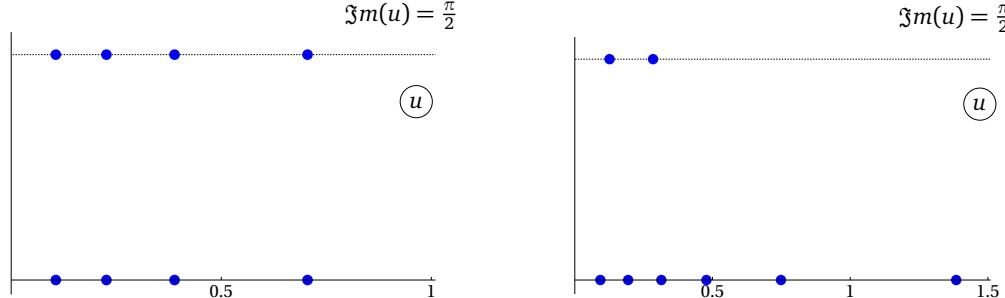

Figure 4: On the left and right panels displayed are the Bethe roots in the complex $u = -\frac{1}{2}\log(\zeta)$ plane for the ground state and an excited state, respectively, of the spin chain with $L = 10$. The excitation is built by disbalancing the number of roots on the two lines with respect to the ground state pattern. The states have total $U_q\big(\mathfrak{sl}(2)\big)$ spin $\mathcal{S} = 2$ while $n = 2.9$.

where $\Delta$ is the conformal dimension, while $c$ stands for the central charge. One finds that

$$\Delta - \frac{c}{24} = \frac{p^2}{n+2} + \frac{s^2}{n} - \frac{1}{12} + \mathtt{d} \tag{3.13}$$

with $\mathtt{d} = 0$. Thus we conclude that the spectrum of scaling dimensions develops a continuous component labeled by the parameter $s \in \mathbb{R}$.

It should be mentioned that in the work [23] RG trajectories were also constructed for which $b(L)$ tends to a pure imaginary number in the scaling limit. Among them is the low energy state mentioned in the paragraph containining formula (3.8), whose typical pattern of Bethe roots in the complex $u = -\frac{1}{2}\log(\zeta)$ plane is depicted in Fig. 3. Moreover, the authors of ref. [23] propose that the imaginary values of $s$ that appear in the scaling limit of the lattice model must satisfy the condition

$$s = \pm\mathrm{i}\Big(-p - \frac{1}{2} - a\Big) \qquad \text{with} \qquad a = 0, 1, 2, \ldots < -p - \tfrac{1}{2} \qquad (\mathtt{d} = 0). \tag{3.14}$$

## 3.2   ODE/IQFT correspondence

In order to characterize the scaling limit of a lattice system, where the spectrum of scaling dimensions possesses a continuous component, mathematical techniques are required that go beyond those used in the works [22, 23]. One of these is an approach to the study of the scaling limit that is based on the so-called ODE/IQFT correspondence [41–44]. We found that the ODEs which describe the scaling limit of the staggered six-vertex model with $U_q\big(\mathfrak{sl}(2)\big)$ invariant BCs lie within the class of differential equations considered in refs. [15, 26] in the context of the lattice system with quasi-periodic BCs. This made the analysis performed in our work possible. In addition, it allowed us to transfer over many previous results concerning the differential equations. Here we briefly discuss the application of the ODE/IQFT correspondence to the study of the scaling limit of the spin chain, while referring the reader to ref. [15] for technical details.

The primary Bethe states, i.e., the RG trajectories where the energy obeys the asymptotic formula (3.3) with $\mathtt{d} = 0$, are labeled by the $U_q\big(\mathfrak{sl}(2)\big)$ spin $\mathcal{S}$ and the RG invariant $s$ defined in eq. (3.11). The ODE/IQFT correspondence implies a relation between the scaling limit of the eigenvalue of the $Q$ operator for $|\Psi_L\rangle$ and the spectral determinant of the ODE:

$$\left[ -\frac{\mathrm{d}^2}{\mathrm{d}z^2} + \frac{p^2 - \frac{1}{4}}{z^2} + \frac{2\mathrm{i}s}{z} + 1 + \mu^{-2-n}\, z^n \right]\psi = 0 \,. \tag{3.15}$$

Here $p$ is given in terms of $\mathcal{S}$ as in eq. (3.5), while $n > 0$ is related to the anisotropy parameter of the spin chain, see (3.2). In addition, in taking the scaling limit the parameter $\zeta$ entering into the $Q$ operator is assigned a certain $L$ dependence such that $\zeta \propto L^{-\frac{n}{n+2}}$. Then $\mu$ appearing in the ODE (3.15) is given by

$$\mu = -\mathrm{i}\left(L/L_0\right)^{\frac{n}{n+2}} \zeta \tag{3.16}$$

with

$$L_0 = \frac{\sqrt{\pi}\,\Gamma\left(1 + \frac{1}{n}\right)}{4\Gamma\left(\frac{3}{2} + \frac{1}{n}\right)}. \tag{3.17}$$

The spectral determinant $D(\mu\,|\,p,s)$ is defined in the following way. One specifies a solution to the differential equation by its behaviour in the vicinity of the singular point $z = 0$:

$$\psi_p(z) \to z^{\frac{1}{2}+p} \qquad \text{as} \qquad z \to 0. \tag{3.18}$$

For large $z$ the term $\mu^{-2-n} z^n$ in (3.15) becomes dominant and one can define another solution through the $z \to +\infty$ asymptotic:

$$\Xi(z) \asymp \left(\frac{z}{\mu}\right)^{-\frac{n}{4}} \exp\left[-\frac{2}{n+2}\left(\frac{z}{\mu}\right)^{\frac{n}{2}+1} {}_2F_1\left(-\frac{1}{2}, -\frac{n+2}{2n}, \frac{n-2}{2n}\,\Big|-\mu^{n+2} z^{-n}\right) + o(1)\right] \tag{3.19}$$

with ${}_2F_1(a, b, c|z)$ being the usual Gauss hypergeometric function and we make the technical assumption that $\mu > 0$ and $n \neq \frac{2}{2k-1}$ with $k = 1, 2, \ldots$ . The spectral determinant $D(\mu) = D(\mu\,|\,p,s)$ is given by

$$D(\mu) = \sqrt{\pi}\,(n+2)^{-\frac{2p}{n+2}-\frac{1}{2}}\,\mu^{-p+\frac{1}{2}}\,\frac{W[\psi_p, \Xi]}{\Gamma\left(1 + \frac{2p}{n+2}\right)}, \tag{3.20}$$

where $W[\psi_p, \Xi] = \Xi\,\partial_z \psi_p - \psi_p\,\partial_z \Xi$ is the Wronskian. The overall factor has been chosen to ensure the normalization

$$D(0) = 1. \tag{3.21}$$

It should be mentioned that the procedure for computing the spectral determinant based on formula (3.20) with the solutions $\psi_p$ and $\Xi$ obtained via a numerical integration of the ODE (3.15) works literally only for $\Re e(p) \geq 0$. Nevertheless $D(\mu\,|\,p,s)$ turns out to be a meromorphic function of $p$ and can be defined for generic complex values of this variable via analytic continuation.

Rather than considering the eigenvalue of $\mathbb{Q}^{(0)}(\zeta)$ (2.24) for a primary Bethe state, we instead discuss the scaling limit of

$$A(\zeta) = \prod_{j=1}^{L-\mathcal{S}} (1 - \zeta/\zeta_j)(1 - \zeta\zeta_j). \tag{3.22}$$

It does not involve the overall factor $\zeta^{\mathcal{S}}$ and the normalization has been imposed such that $A(0) = 1$. Then, the scaling relation between $A(\zeta)$ and the spectral determinant reads as

$$\operatorname*{slim}_{\substack{L\to\infty \\ b(L)\to s}} G^{(L)}\left(-\mu^2 \,\Big|\, \tfrac{2}{n+2}\right) A\left(\left(L/L_0\right)^{-\frac{n}{n+2}} \mathrm{i}\mu\right) = D(\mu). \tag{3.23}$$

Here the function $G$ has been chosen to ensure the convergence of the limit and is given by

$$G^{(L)}(E\,|\,g) = \exp\left(\sum_{m=1}^{\left[\frac{1}{2(1-g)}\right]} \frac{(-1)^m L}{m\cos(\pi m g)}\left(\frac{L}{L_0}\right)^{2m(g-1)} E^m\right), \tag{3.24}$$

(this should be compared with formula (5.48) in the work [15]), where the brackets [...] stand for the integer part, while $L_0$ is the same as in (3.17). As above, the technical assumption

$$n \neq \frac{2}{2k-1} \tag{3.25}$$

with $k = 1, 2, \ldots$ is being made, see ref. [15] for details concerning the case $n = \frac{2}{2k-1}$.

For the RG trajectories, where d entering into the asymptotic formula for the energy (3.3) is greater than zero, the scaling relation (3.23) is modified as follows. The l.h.s. remains the same, while for the r.h.s. one takes $D(\mu)$ to be the spectral determinant for the differential equation:

$$\left[ -\frac{d^2}{dz^2} + \frac{p^2 - \frac{1}{4}}{z^2} + \frac{2is}{z} + 1 + \sum_{a=1}^{d} \left( \frac{2}{(z-w_a)^2} + \frac{n}{z(z-w_a)} \right) + \mu^{-2-n} z^n \right] \psi = 0 . \tag{3.26}$$

Here

$$\boldsymbol{w} = (w_1, \ldots, w_d) \tag{3.27}$$

are not arbitrary parameters. They are restricted by the condition that any solution $\psi(z)$ of the differential equation must be single valued in the vicinity of $z = w_a$. This leads to the coupled algebraic system:

$$4n\,w_a^2 \;+\; 8is\,(n+1)\,w_a - (n+2)\left((n+1)^2 - 4p^2\right) \tag{3.28}$$

$$+\; 4 \sum_{b \neq a}^{d} \frac{w_a\left((n+2)^2\,w_a^2 - n(2n+5)\,w_a w_b + n(n+1)\,w_b^2\right)}{(w_a - w_b)^3} = 0 \qquad (a = 1, \ldots, d).$$

For generic $n$, $s$ and $p$ the number of solutions $\boldsymbol{w} = \{w_a\}_{a=1}^{d}$, up to permutations of the $w_a$'s, is given by $\mathrm{par}_2(d)$ – the number of bipartitions of d. The generating function for this combinatorial quantity reads as:

$$\sum_{d=0}^{\infty} \mathrm{par}_2(d)\,q^d = \prod_{j=1}^{\infty} \frac{1}{(1-q^j)^2} . \tag{3.29}$$

For applications to the staggered six-vertex model with $U_q\big(\mathfrak{sl}(2)\big)$ invariant BCs $p$ is not generic, but should be taken as in (3.5), i.e., $2p = 2\mathcal{S} + 1 - (n+2)$. Then, it turns out that the number of solutions of the coupled equations (3.28) is typically less than $\mathrm{par}_2(d)$. To explain this phenomenon, let's replace $p$ with $p_\varepsilon = p + \varepsilon^{2\mathcal{S}+1}$ where $0 < \varepsilon \ll 1$. Of the $\mathrm{par}_2(d)$ solution sets of (3.28) with $p \mapsto p_\varepsilon$ there exist those where

$$w_a = O(\varepsilon) \qquad \text{for} \qquad a = 1, 2, \ldots, 2\mathcal{S} + 1 . \tag{3.30}$$

The other variables $\{w_a\}_{a=2\mathcal{S}+2}^{d}$ tend to a finite, nonvanishing limit as $\varepsilon \to 0$. Their limiting values obey (3.28) with the replacements $p \mapsto \mathcal{S} + \frac{1}{2} + \frac{1}{2}(n+2)$ and $d \mapsto d - 2\mathcal{S} - 1$. In counting the solution sets of the algebraic system on $w_a$ with $2p = 2\mathcal{S} + 1 - (n+2)$ we only consider those to be admissible where none of the $w_a$ are zero. It is easy to see that

$$N(d|\mathcal{S}) := \# \text{ of solution sets of (3.28) with } p \text{ as in (3.5)} = \mathrm{par}_2(d) - \mathrm{par}_2(d - 2\mathcal{S} - 1) \tag{3.31}$$

(we take by definition $\mathrm{par}_2(d) = 0$ when its argument is a negative integer).

We suppose that for a given trajectory $\{|\Psi_L\rangle\}$ with RG invariants $\mathcal{S}$, $s$ and d there exists a solution set $\boldsymbol{w}$ of (3.28) such that the scaling relation (3.23) holds true with $D(\mu) = D(\mu\,|\,\boldsymbol{w}, p, s)$

being the spectral determinant for the differential equation (3.26). Note that eqs. (3.18)-(3.20) for the definition of $D(\mu)$ still remain valid since the inclusion of the extra sum in the ODE has no impact on the leading asymptotics of $\psi_p$ and $\Xi$.

Unfortunately, we do not know of a way of rigorously proving the above statement. We checked it numerically for a variety of cases using the so-called sum rules. The analysis is not included here, as it is essentially the same as that presented in section 11 of the work [15] concerning the staggered six-vertex model with quasi-periodic BCs. We wish, however, to mention a scaling relation involving the products over the Bethe roots,

$$\Pi_{\pm} = \prod_{m=1}^{L-S} q\left(\zeta_m \pm \mathrm{i}q^{-1}\right)\left(\zeta_m^{-1} \pm \mathrm{i}q^{-1}\right), \tag{3.32}$$

that will become important later. It involves the coefficients $\mathfrak{C}_{p,s}^{(\pm)} = \mathfrak{C}_{p,s}^{(\pm)}(\boldsymbol{w})$, which occur in the large $\mu$ asymptotic expansion of $D(\mu)$:

$$D(\mu\,|\,\boldsymbol{w},p,s) \asymp \mathfrak{C}_{p,s}^{(\pm)}(\boldsymbol{w})\,(\pm\mu)^{\pm\frac{\mathrm{i}(n+2)s}{n}-p}\,\exp\left(\frac{2L_0}{\cos(\frac{\pi}{n})}\,(\pm\mu)^{\frac{n+2}{n}} + o(1)\right) \quad \text{for} \quad \Re e(\pm\mu) > 0 \tag{3.33}$$

(again, we assume that $n \neq \frac{2}{2k-1}$ with $k = 1, 2, \dots$). For the case when $\mathtt{d} = 0$, the coefficients are given by

$$\mathfrak{C}_{p,s}^{(0,\pm)} = \sqrt{\frac{2\pi}{n+2}}\;2^{-p\pm\frac{\mathrm{i}(n+2)s}{n}}\,(n+2)^{-\frac{2p}{n+2}}\,\frac{\Gamma(1+2p)}{\Gamma(1+\frac{2p}{n+2})\,\Gamma(\frac{1}{2}+p\pm\mathrm{i}s)}\;. \tag{3.34}$$

In general,

$$\mathfrak{C}_{p,s}^{(\pm)}(\boldsymbol{w}) = \mathfrak{C}_{p,s}^{(0,\pm)}\,\check{\mathfrak{c}}_{p,s}^{(\pm)}(\boldsymbol{w}), \tag{3.35}$$

where $\check{\mathfrak{c}}^{(\pm)}$ are normalized to be one for $\mathtt{d} = 0$. A closed form expression for $\check{\mathfrak{c}}^{(\pm)}$ for general $\mathtt{d} = 0, 1, 2, \dots$ was obtained in ref. [45] and, for the reader's convenience, is reproduced in this paper in Appendix A. The scaling relation reads as

$$\Pi_{\pm} \asymp \frac{C}{2\cos(\frac{\pi}{n+2})}\,\mathrm{e}^{\pm\frac{\pi}{n}s}\,\mathfrak{C}_{p,s}^{(\pm)}(\boldsymbol{w})\left(\frac{L}{L_0}\right)^{-\frac{np}{n+2}\pm\mathrm{i}s}\left(\frac{4n}{n+2}\right)^{L}\left(1+O(L^{-\epsilon})\right). \tag{3.36}$$

It involves a non-universal constant $C$, which is expressed in terms of

$$\hat{\tau}(\omega) = -\frac{1}{4\pi}\,\frac{\sinh(\frac{\pi(n-1)}{4(n+2)}\,\omega)}{\sinh(\frac{\pi\omega}{4(n+2)})\cosh(\frac{n\pi\omega}{4(n+2)})} \tag{3.37}$$

and the Lerch transcendent

$$\Phi(z,s,a) = \sum_{m=0}^{\infty}\frac{z^m}{(m+a)^s}\;. \tag{3.38}$$

Namely,

$$C = \exp\left(2\int_{-\infty}^{\infty}\mathrm{d}\omega\left(\frac{\hat{\tau}(\omega)}{\omega}\left(\Im m\left[\mathrm{e}^{\frac{2\mathrm{i}\pi}{n+2}}\,\Phi\left(\mathrm{e}^{-\frac{\mathrm{i}n\pi}{n+2}}, 1, 1-\frac{\mathrm{i}\omega}{4}\right)\right] - \frac{\pi}{n+2} - \frac{2}{\omega}\right) - \frac{n-1}{2\pi\omega^2}\right)\right). \tag{3.39}$$

Also, the notation $O(L^{-\epsilon})$ with some $\epsilon > 0$ means that the correction terms fall off faster than any power of the logarithm of $L$. The asymptotic formula (3.36) is the analogue for the lattice system with open $U_q(\mathfrak{sl}(2))$ invariant BCs of a product rule presented in the work [15], see (11.19) therein.

## 3.3 Quantization condition

Consider the problem of computing the spectrum of conformal dimensions of the critical lattice system and the classification of the space of states $\mathcal{H}$ appearing in the scaling limit. The asymptotic relation for the energy (3.3), together with the numerical analysis from refs. [22, 23] concerning the large $L$ behaviour of $b(L)$ for a class of states, shows that $\mathcal{H}$ contains two components. There is a 'continuous component', where $s = \text{slim}_{L\to\infty}\, b(L)$ parameterising the conformal dimensions as in eq. (3.13) may take any real value, as well as a 'discrete' one for which $s$ belongs to a finite set of pure imaginary numbers. In the next section a full description of these linear spaces will be given. Among other things, this includes the admissible values of pure imaginary $s$ as well as the density of states characterizing the continuous spectrum. The results are based on an analysis of the so-called 'quantization condition' for $b(L)$, which we shall obtain below.

The key observation is that the square root of the eigenvalue of the quasi-shift operator (3.7), used in the computation of $b(L)$ (3.6), may be expressed as

$$\sqrt{B} = (-1)^{L-\mathcal{S}}\, \frac{\Pi_+}{\Pi_-} \,. \tag{3.40}$$

Here $\Pi_\pm$ stand for the products over the Bethe roots defined in (3.32). Let us substitute these products for their asymptotics (3.36) with $s$ replaced by the 'running coupling' $b(L)$. Upon rearranging and making use of eq. (3.6), one finds

$$\left(\frac{L}{L_0}\right)^{2is} e^{\frac{i}{2}\delta(\boldsymbol{w},p,s)}\Bigg|_{s=b(L)} = \sigma + O(L^{-\epsilon}) \tag{3.41a}$$

with

$$e^{\frac{i}{2}\delta(\boldsymbol{w},p,s)} = \frac{\mathfrak{C}^{(+)}_{p,s}(\boldsymbol{w})}{\mathfrak{C}^{(-)}_{p,s}(\boldsymbol{w})} \tag{3.41b}$$

and

$$\sigma = (-1)^{L-\mathcal{S}} \,. \tag{3.41c}$$

The relation (3.41), which will be henceforth referred to as the quantization condition, is interpreted in the following way. Given an RG trajectory $\{|\Psi_L\rangle\}$ one computes $p$ from the value of the $U_q\big(\mathfrak{sl}(2)\big)$ spin $\mathcal{S}$ via the definition (3.5) as well as the sign factor $\sigma = (-1)^{L-\mathcal{S}}$. The latter is kept fixed along $\{|\Psi_L\rangle\}$ since, in the construction of RG trajectories, $L$ is always increased by two (see section 2). Then $b(L)$ computed from the Bethe roots according to eqs. (3.7) and (3.6) obeys (3.41) for some solution set $\boldsymbol{w} = \{w_a\}_{a=1}^{\mathrm{d}}$ of the algebraic system (3.28) with $s$ replaced by $b(L)$. The 'phase shift' $\delta$ is given in terms of the coefficients $\mathfrak{C}^{(\pm)}_{p,s}(\boldsymbol{w})$ (3.35), which were introduced in the previous subsection. For the primary Bethe states with $\mathrm{d} = 0$, one has

$$e^{\frac{i}{2}\delta(\emptyset,p,s)} = 2^{\frac{2i(n+2)s}{n}}\, \frac{\Gamma(\frac{1}{2}+p-is)}{\Gamma(\frac{1}{2}+p+is)} \qquad (\mathrm{d}=0). \tag{3.42}$$

For $\mathrm{d} = 1,2,3,\dots$ one must make use of eqs. (3.34), (3.35) together with the explicit formula for $\breve{\mathfrak{C}}^{(\pm)}_{p,s}(\boldsymbol{w})$ as a function of $p$, $s$ and $\boldsymbol{w} = \{w_a\}_{a=1}^{\mathrm{d}}$ contained in Appendix A.

Let's take a moment to discuss the quantization condition (3.41) for the primary Bethe states in the context of the results of the previous works [22, 23]. We start with the asymptotic (3.10) for $b(L)$ that was observed for a class of RG trajectories labelled by the integer $\mathfrak{m}$. In this case, it is useful to take the logarithm of both sides of formula (3.41) with the phase shift as in (3.42) and write it in the form:

$$2b_{\mathfrak{m}}\log\left(\frac{L}{L_0}\right) - i\log\left[2^{\frac{2i(n+2)b_{\mathfrak{m}}}{n}}\, \frac{\Gamma(\frac{1}{2}+p-ib_{\mathfrak{m}})}{\Gamma(\frac{1}{2}+p+ib_{\mathfrak{m}})}\right] = \pi\mathfrak{m} + O(L^{-\epsilon}) \,. \tag{3.43}$$

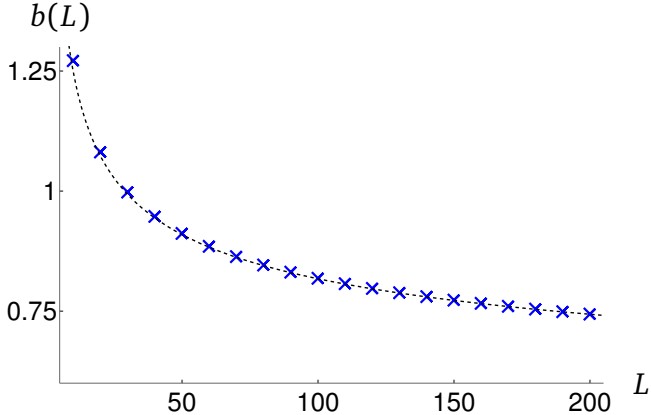

Figure 5: The numerical data comes from an RG trajectory $\{|\Psi_L\rangle\}$, where the representative state for $L = 10$ is the one whose pattern of Bethe roots is displayed on the right panel of fig. 4. In particular, it has $\mathtt{d} = 0$; total $U_q\big(\mathfrak{sl}(2)\big)$ spin $\mathcal{S} = 2$; while the integer $\mathfrak{m}$ — the difference between the number of roots lying on the real line and the line $\mathfrak{Im}(u) = \frac{\pi}{2}$ in the complex $u$ plane — is held fixed along the flow to be $\mathfrak{m} = 4$. The crosses depict the numerical values of $b(L) = b_\mathfrak{m}(L)$ for different $L$ which were computed from the Bethe roots corresponding to $|\Psi_L\rangle$ via formulae (3.6) and (3.7). The dashed line gives the predictions coming from the quantization condition (3.43) with $\mathfrak{m} = 4$ and the correction terms ignored. Note that the branch of the logarithm was fixed by requiring that the l.h.s. of (3.43) is a continuous function for real $b_\mathfrak{m}$ which vanishes at $b_\mathfrak{m} = 0$.

For the class of states we are considering $b_\mathfrak{m}(L)$ goes to zero as $L \to \infty$. As a result, the second term in the l.h.s. of the above relation containing the $\Gamma$-functions also tends to zero and, to a first approximation, can be ignored. This way one obtains (3.10). Formula (3.43) provides a refinement to the large $L$ asymptotic behaviour of $b_\mathfrak{m}(L)$ which takes into account all power law corrections in $1/\log(L)$. To demonstrate its accuracy, some numerical data obtained from the Bethe roots for a primary Bethe state $|\Psi_L\rangle$ is compared with the predictions coming from the quantization condition in Fig. 5.

Another possibility of how (3.41) could be satisfied for $L \gg 1$ is if $b(L)$ approaches a singularity of the phase shift. The explicit formula (3.42), valid for $\mathtt{d} = 0$, shows that these occur for pure imaginary $s$ when $\frac{1}{2} + p \pm \mathrm{i}s$ is a positive integer. If the imaginary part of $b(L)$ is positive, then the vanishing of the first term in the l.h.s. of (3.41) may be compensated if $b(L)$ tends to a pole of $\mathrm{e}^{\frac{\mathrm{i}}{2}\delta}$, i.e.,

$$\operatorname*{slim}_{L\to\infty} b(L) = s \qquad \text{with} \qquad s = \mathrm{i}\big(-p - \tfrac{1}{2} - \ell\big) \qquad \big(\mathfrak{Im}(b(L)) > 0\big) \qquad (3.44)$$

and $\ell = 0, 1, 2, \ldots$ . This is the same as eq. (3.14) with the sign factor chosen to be '+'. The upper bound on $\ell$ in that equation ensures the condition $\mathfrak{Im}(b(L)) > 0$. The minus version of the relation is deduced from (3.41) by means of similar arguments.

A verification of the quantization condition (3.41) was carried out using numerical data obtained from the lattice model with $L = 10$. The spin chain Hamiltonian was constructed and the first few hundred lowest energy Bethe states were found via a direct diagonalization procedure. Note that, because of the $U_q\big(\mathfrak{sl}(2)\big)$ symmetry, it was sufficient to focus on the sector with $S^z = 0$ as there is always one state $|\Psi_L\rangle$ from the $U_q\big(\mathfrak{sl}(2)\big)$ multiplet $\mathcal{M}_\mathcal{S}$ lying in this sector. For each Bethe state, apart from the energy, the eigenvalue of the quasi-shift operator was computed from which we extracted $b$. The numerical data for $b(L)$ was compared with $b_*(L)$ — the predictions coming from the quantization condition. The latter was obtained by considering (3.41) with $L = 10$ and the correction term $O(L^{-\epsilon})$ ignored as an equation from

which $b_*(L)$ could be determined numerically. Note that the phase shift $e^{\frac{i}{2}\delta}$ therein depends on $b$ transcendentally via the Gamma functions as in (3.42) and algebraically through the set $\boldsymbol{w}$, which solves the coupled system (3.28) with $s \mapsto b$. For given $\mathrm{d} \leq 3$ we took the $\mathrm{par}_2(\mathrm{d}) - \mathrm{par}_2(\mathrm{d} - 2\mathcal{S} - 1)$ equations which are obtained from the quantization condition by specializing the phase shift $\delta = \delta(\boldsymbol{w}, p, s)$ to different solution sets $\boldsymbol{w} = \{w_a\}_{a=1}^{\mathrm{d}}$ of (3.28). For each of them we found all possible solutions $b_*(L)$ that lie in a suitably chosen finite portion of the strip $\left|\Im m(b_*)\right| < \frac{n}{2} + \varepsilon$ with $0 < \varepsilon \ll 1$ of the complex $b$ plane. Some of the results for the comparison of $b(L)$ and $b_*(L)$ for $L = 10$ are presented in Fig. 6. They motivated us to make the following conjecture.

**Conjecture:** For any RG trajectory $\{|\Psi_L\rangle\}$ labeled by $\mathcal{S}$, $\mathrm{d}$ and a solution set $\boldsymbol{w} = \{w_a\}_{a=1}^{\mathrm{d}}$ of eq. (3.28) the corresponding value of $b(L) = \frac{n}{2\pi} \log\left(\sqrt{B}\right)$, with $\sqrt{B}$ computed according to formula (3.7), obeys the quantization condition (3.41). Conversely, let $b_*(L)$ with $-\frac{n}{2} < \Im m\left(b_*(L)\right) \leq \frac{n}{2}$ be a solution of the relation (3.41) with the correction terms ignored. Then, there exists a unique $U_q\left(\mathfrak{sl}(2)\right)$ multiplet $\mathcal{M}_{\mathcal{S}}$ for which $\sqrt{B}$ obtained from $|\Psi_L\rangle \in \mathcal{M}_{\mathcal{S}}$ is such that $\sqrt{B} - \exp\left(\frac{2\pi}{n} b_*(L)\right)$ goes to zero faster than any power of the logarithm of $L$.

# 4 Space of states in the scaling limit

## 4.1 Continuous and discrete spectrum

A key result of this paper is the conjecture, above, which was motivated by our numerical work. It describes a certain one-to-one relation between $b(L)$ and $b_*(L)$. The former is obtained via the Bethe roots corresponding to a state $|\Psi_L\rangle$ in a multiplet $\mathcal{M}_{\mathcal{S}}$, labeled by $2p = 2\mathcal{S}+1-(n+2)$, the non-negative integer $\mathrm{d}$ and one of the $N(\mathrm{d}\,|\,\mathcal{S}) = \mathrm{par}_2(\mathrm{d}) - \mathrm{par}_2(\mathrm{d} - 2\mathcal{S} - 1)$ solution sets $\boldsymbol{w} = \{w_a\}_{a=1}^{\mathrm{d}}$ of the algebraic system (3.28). Since it requires the construction of an RG trajectory, computing $b(L)$ for $L \gg 1$ can be cumbersome and time-consuming to carry out. The notation $b_*(L)$ stands for a solution of the quantization condition (3.41) treated as an equation for $b(L)$ with the correction terms ignored. Accepting the conjecture to be true, one can determine the spectrum of $b(L)$ for the low energy states at large $L$ via a study of (3.41). This is extremely powerful, since much less computing resources are needed to numerically analyze the quantization condition and it may be studied analytically as well. The results allow one to characterize the spectrum of conformal dimensions together with the space of low energy states in the scaling limit.

Consider the quantization condition (3.41) and suppose that $\delta(\boldsymbol{w}, p, s)|_{s \mapsto b(L)} \ll \log(L)$. Then the first term dominates and one can develop an asymptotic expansion for $b(L)$ in $1/\log(L)$. The leading and subleading asymptotic behaviour reads as

$$b_{\mathfrak{m}}(L) = \frac{2\pi \mathfrak{m} - \delta_0}{4 \log\left(e^{\frac{1}{4}\delta_0'} L/L_0\right)} + O\left((\log L)^{-3}\right) \qquad \left(L \gg 1, \; \mathfrak{m} - \text{fixed}\right). \qquad (4.1)$$

Here we use the notation

$$\delta_0 = \delta|_{s=0}, \qquad\qquad \delta_0' = \partial_s \delta|_{s=0}, \qquad\qquad (4.2)$$

while $\mathfrak{m}$, which labels the different $b(L)$ obeying the quantization condition, comes about as a result of taking the logarithm of (3.41) and is an even or odd integer for $\sigma = +1$ or $\sigma = -1$, respectively. Formula (4.1) shows that, in general, $b_{\mathfrak{m}}(L)$ is a complex number since $\delta_0$ and $\delta_0'$ are generically complex. However, while $\Im m\left(b_{\mathfrak{m}}(L)\right) \sim 1/\log(L) \to 0$ as

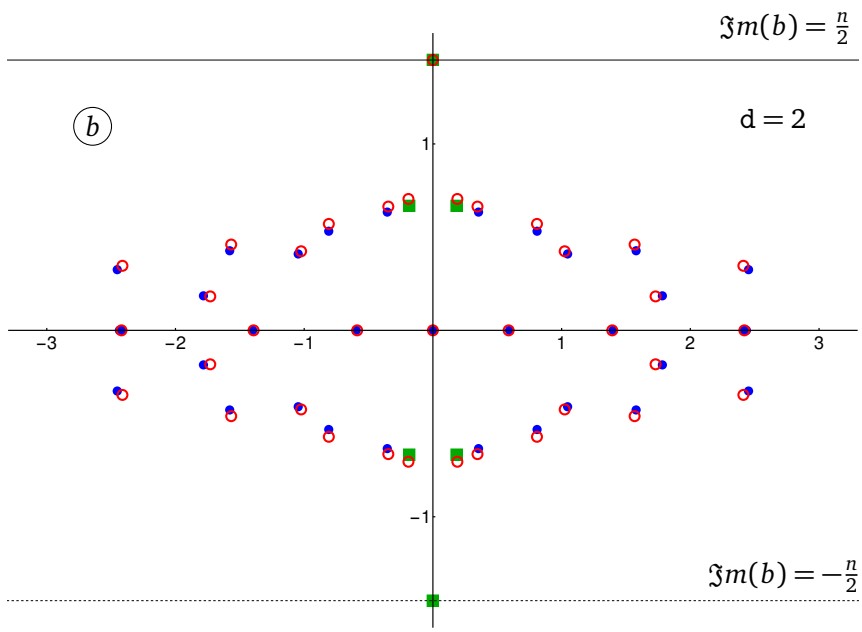

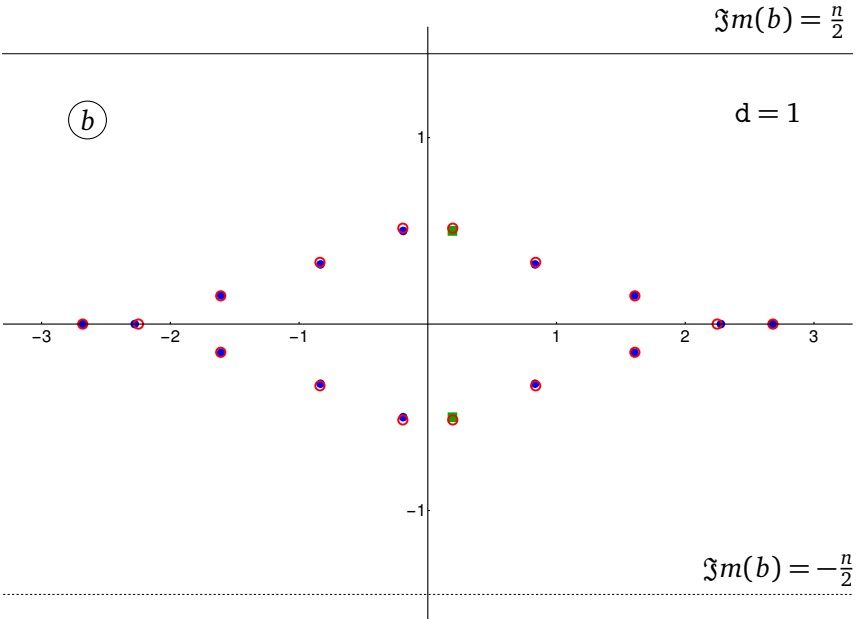

Figure 6: Numerical data for $b(L)$ and $b_*(L)$ is plotted in the complex $b$ plane for the lattice system with $L = 10$. Out of the few hundred states that were considered, only those with $\mathtt{d} = 2$ (top panel), $\mathtt{d} = 1$ (bottom panel) and $U_q\big(\mathfrak{sl}(2)\big)$ spin $\mathcal{S} = 2$ were used to produce the figure. The open circles correspond to $b(L)$ that was extracted from the Bethe roots by means of eqs. (3.7), (3.6). The filled shapes represent $b_*(L)$ obtained from an analysis of the quantization condition (3.41). The green squares and blue circles are used to distinguish whether in the scaling limit $b_*(L)$ becomes a pure imaginary number or a real number, respectively. The two green squares in the top panel for which $\mathrm{slim}_{L\to\infty} b_*(L) = \pm\frac{\mathrm{i}n}{2}$ correspond to the same state. It seems interesting to note that the agreement between $b(L)$ and $b_*(L)$ is better than in the case of the lattice model with quasi-periodic boundary conditions imposed, compare the above figures with the ones contained in Appendix C of ref. [15]. The parameter $n$ was taken to be $n = 2.9$.

$L \to \infty$ the magnitude of the real part is controlled by the integer $\mathfrak{m}$ which, for the low energy states, may take any values up to some $\mathfrak{m}_{\max} \ll L$. Numerical work leads us to suppose that $\lim_{L \to \infty} b_{\mathfrak{m}_{\max}} = \infty$.

Let $\mathcal{H}^{(\mathrm{cont})}_{L|\mathcal{S}}$ denote the set of low energy states $|\Psi_L\rangle$ with fixed value of the $U_q(\mathfrak{sl}(2))$ spin $\mathcal{S} = 0, 1, 2, \ldots$ such that $\mathfrak{Im}(b(L)) \to 0$ as $L \to \infty$. Recall that the states come in multiplets $\mathcal{M}_{\mathcal{S}}$ and one can choose a basis for that multiplet in which the $z$-projection of the total spin, $\mathbb{S}^z = \frac{1}{2} \sum_{J=1}^{2L} \sigma_J^z$, is diagonal. This yields the refinement

$$\mathcal{H}^{(\mathrm{cont})}_{L|\mathcal{S}} = \bigcup_{S^z=-\mathcal{S}}^{\mathcal{S}} \mathcal{H}^{(\mathrm{cont})}_{L|\mathcal{S},S^z} . \tag{4.3}$$

Each low energy Bethe state in $\mathcal{H}^{(\mathrm{cont})}_{L|\mathcal{S},S^z}$ is uniquely specified by the non-negative integer $\mathrm{d}$, a solution set $\boldsymbol{w} = \{w_a\}_{a=1}^{\mathrm{d}}$ of the algebraic system (3.28) with $s \mapsto b(L)$, as well as the even or odd integer $\mathfrak{m}$ that enters into the asymptotics (4.1). For $L \gg 1$ the value of $b_{\mathfrak{m}}(L)$ becomes densely distributed in the segment $(-b_{\mathfrak{m}_{\max}}(L), +b_{\mathfrak{m}_{\max}}(L))$. The density of states is obtained from the quantization condition (3.41) written in logarithmic form:

$$\left[ 4s \log(L/L_0) + \delta(\boldsymbol{w}, p, s) \right]|_{s \mapsto b_{\mathfrak{m}}(L)} = 2\pi\mathfrak{m} + O(L^{-\epsilon}) . \tag{4.4}$$

Here the branch of the logarithm needed to define $\delta$ from the relation (3.41b) is taken such that the phase shift is a continuous function of $s$ in the strip $|\mathfrak{Im}(s)| < \varepsilon$ for some $\varepsilon > 0$ (it is being assumed that $\mathrm{e}^{\frac{\mathrm{i}}{2}\delta}$ contains no zeroes or poles for real $s$). The term in the square brackets in the left hand side of (4.4) is a monotonic function of $s$ for $L$ sufficiently large. This way one concludes that the number of states in $\mathcal{H}^{(\mathrm{cont})}_{L|\mathcal{S},S^z}$ with fixed $\mathrm{d}$ such that $\mathfrak{Re}(b(L))$ lies in the interval $(s, s + \Delta s) \in (-b_{\mathfrak{m}_{\max}}(L), +b_{\mathfrak{m}_{\max}}(L))$ is given by $\rho_p^{(\mathrm{d})}(s) \Delta s$ with[6]

$$\rho_p^{(\mathrm{d})}(s) = \frac{1}{\pi} N(\mathrm{d}|\mathcal{S}) \log\left(2^{\frac{n+2}{n}} L/L_0\right) + \frac{1}{2\pi\mathrm{i}} \, \partial_s \log\left[ \left( \frac{\Gamma(\frac{1}{2} + p - \mathrm{i}s)}{\Gamma(\frac{1}{2} + p + \mathrm{i}s)} \right)^{N(\mathrm{d}|\mathcal{S})} \prod_{\substack{\boldsymbol{w} \\ \mathrm{d-fixed}}} \frac{\check{\mathfrak{c}}_{p,s}^{(+)}(\boldsymbol{w})}{\check{\mathfrak{c}}_{p,s}^{(-)}(\boldsymbol{w})} \right] \tag{4.5}$$

up to corrections which vanish as $L \to \infty$. The product over $\boldsymbol{w}$ appearing in the r.h.s. goes over all the $N(\mathrm{d}|\mathcal{S})$ (3.31) solution sets of the algebraic system (3.28) with $\mathrm{d}$ fixed. Also, recall that $2p = 2\mathcal{S} + 1 - (n+2)$.

In the work [45] a formula is presented for a product over $\boldsymbol{w}$ similar to the one appearing in the r.h.s. of (4.5) (see also Appendix B of [15]). It is valid for the case of generic $p$ and $n$ when the number of solution sets $\boldsymbol{w}$ of (3.28) is $\mathrm{par}_2(\mathrm{d})$. Based on this, one can derive the result:

$$\prod_{\substack{\boldsymbol{w} \\ \mathrm{d-fixed}}} \frac{\check{\mathfrak{c}}_{p,s}^{(+)}(\boldsymbol{w})}{\check{\mathfrak{c}}_{p,s}^{(-)}(\boldsymbol{w})} = (-1)^{\mathrm{par}_2(\mathrm{d}-2\mathcal{S}-1)} \prod_{a=0}^{\mathrm{d}-1} \left( \frac{\frac{1}{2} + a + p - \mathrm{i}s}{\frac{1}{2} + a + p + \mathrm{i}s} \right)^{N(\mathrm{d}|\mathcal{S}) - N_a^+(\mathrm{d}|\mathcal{S})}$$

$$\times \prod_{a=0}^{\mathrm{d}-1} \left( \frac{\frac{1}{2} + a - p - \mathrm{i}s}{\frac{1}{2} + a - p + \mathrm{i}s} \right)^{N(\mathrm{d}|\mathcal{S}) - N_a^-(\mathrm{d}|\mathcal{S})} \tag{4.6}$$

---

[6]This line of arguments is analogous to the standard derivation in the root density approach. One introduces a monotonic increasing counting function which evaluates to (half-)integers at the Bethe roots similar as the l.h.s of (4.4) evaluates to odd/even integers $\mathfrak{m}$ multiplied by $2\pi$ when $s$ is swapped for $b_{\mathfrak{m}}(L)$ and $L \gg 1$. Differentiating the counting function in the root density approach yields the root density, while we obtain the density of states (4.5) by differentiating (4.4) and dividing by $4\pi$.

with the integers $N_a^{\pm}$ being defined through their generating function as

$$\sum_{\mathsf{d}=0}^{\infty} N_a^{\pm}(\mathsf{d}\,|\,\mathcal{S})\,\mathsf{q}^{\mathsf{d}} = \left(\prod_{j=1}^{\infty}(1-\mathsf{q}^j)^{-2}\right)\sum_{m=0}^{\infty}(-1)^m\left(1-\mathsf{q}^{(1\pm m)(2\mathcal{S}+1)}\right)\mathsf{q}^{ma+\frac{m(m+1)}{2}}\,. \qquad (4.7)$$

Notice that

$$N_a^+(\mathsf{d}\,|\,\mathcal{S}) = N(\mathsf{d}\,|\,\mathcal{S}) - N_{-a-1}^-(\mathsf{d}\,|\,\mathcal{S})\,. \qquad (4.8)$$

The scaling limit of the RG trajectory $L \mapsto |\Psi_L\rangle \in \mathcal{H}_{L|\mathcal{S}}^{(\mathrm{cont})}$ labeled by real $s$, the integers $\mathcal{S}$, $S^z$, $\mathsf{d}$ and the solution set $\boldsymbol{w}$ yields

$$\operatorname*{slim}_{L\to\infty} |\Psi_L\rangle = |\psi_{p,s}^{(S^z)}(\boldsymbol{w})\rangle\,. \qquad (4.9)$$

One can define the linear span $\mathcal{H}_{\mathcal{S}}^{(\mathrm{cont})}$ of all such possible states with fixed $\mathcal{S}$. The above discussion implies that this linear space admits the decomposition

$$\mathcal{H}_{\mathcal{S}}^{(\mathrm{cont})} = \bigoplus_{S^z=-\mathcal{S}}^{\mathcal{S}} \mathcal{H}_{\mathcal{S},S^z}^{(\mathrm{cont})}\,, \qquad (4.10)$$

where each of the spaces $\mathcal{H}_{\mathcal{S},S^z}^{(\mathrm{cont})}$ is isomorphic to $\mathcal{H}_{\mathcal{S},\mathcal{S}}^{(\mathrm{cont})}$ and

$$\mathcal{H}_{\mathcal{S},\mathcal{S}}^{(\mathrm{cont})} = \int_{\mathbb{R}}^{\oplus} \mathrm{d}s \bigoplus_{\mathsf{d}=0}^{\infty} \mathcal{H}_{p,s}^{(\mathrm{cont},\mathsf{d})} \qquad \left(2p = 2\mathcal{S}+1-n-2\right). \qquad (4.11)$$

The components appearing inside the direct sum are finite dimensional such that

$$\dim\left(\mathcal{H}_{p,s}^{(\mathrm{cont},\mathsf{d})}\right) = N(\mathsf{d}\,|\,\mathcal{S})\,. \qquad (4.12)$$

For the low energy states where the value of $\mathfrak{Im}\big(b(L)\big)$ is nonvanishing in the limit $L \to \infty$ so that they do not belong to $\mathcal{H}_{L|\mathcal{S}}^{(\mathrm{cont})}$, one may repeat the similar analysis that was performed in ref. [15] in the context of the staggered six-vertex model with quasi-periodic BCs (see Appendix B therein). Let's denote by $\mathcal{H}_{L|\mathcal{S},S^z}^{(\mathrm{disc})}$ the set of such states $|\Psi_L\rangle$ with given quantum numbers $\mathcal{S}$ and $S^z$. The quantization condition (3.41) implies that the set $\boldsymbol{w}$ and $s = \operatorname{slim}_{L\to\infty} b(L)$ labelling the RG trajectory $\{|\Psi_L\rangle\}$ must be such that

$$\mathrm{e}^{-\frac{\mathrm{i}}{2}\delta(\boldsymbol{w},p,s)} = 0 \qquad \text{if} \qquad \mathfrak{Im}(s) > 0\,, \qquad\qquad \mathrm{e}^{+\frac{\mathrm{i}}{2}\delta(\boldsymbol{w},p,s)} = 0 \qquad \text{if} \qquad \mathfrak{Im}(s) < 0\,. \qquad (4.13)$$

We supplement this with the additional constraint on the imaginary part on $s$:

$$-\frac{n}{2} < \mathfrak{Im}(s) \leq \frac{n}{2}\,. \qquad (4.14)$$

It comes from the inequality (3.8), while the line $\mathfrak{Im}(s) = -\frac{\mathrm{i}n}{2}$ was excluded from the interval in order to avoid double counting states with $\operatorname{slim}_{L\to\infty} b(L) = \pm\frac{\mathrm{i}n}{2}$. It turns out that the phase shift satisfies:

$$\mathrm{e}^{\frac{\mathrm{i}}{2}\delta(\boldsymbol{w},p,s)} = \mathrm{e}^{-\frac{\mathrm{i}}{2}\delta(-\boldsymbol{w},p,-s)}\,. \qquad (4.15)$$

Here $-\boldsymbol{w}$ denotes the set $\{-w_a\}_{a=1}^{\mathsf{d}}$, where if $\boldsymbol{w}$ obeys the algebraic system (3.28) then $-\boldsymbol{w}$ obeys the same set of equations with $s \mapsto -s$. This allows one to focus on the case with $0 < \mathfrak{Im}(s) \leq \frac{n}{2}$, while results for $-\frac{n}{2} < \mathfrak{Im}(s) < 0$ follow by simply flipping the sign $s \mapsto -s$.

The analysis of (4.13) is greatly facilitated by the relation

$$\prod_{\substack{\boldsymbol{w} \\ \mathrm{d-fixed}}} \mathrm{e}^{\frac{\mathrm{i}}{2}\delta(\boldsymbol{w},p,s)} = \left(2^{(2n+4)\frac{\mathrm{i}s}{n}} \frac{\Gamma(\frac{1}{2}+p-\mathrm{i}s)}{\Gamma(\frac{1}{2}+p+\mathrm{i}s)}\right)^{N(\mathrm{d}|\mathcal{S})} \prod_{\substack{\boldsymbol{w} \\ \mathrm{d-fixed}}} \frac{\check{\mathfrak{C}}_{p,s}^{(+)}(\boldsymbol{w})}{\check{\mathfrak{C}}_{p,s}^{(-)}(\boldsymbol{w})} \tag{4.16}$$

with the last term in the r.h.s. being given by the product (4.6). It follows from the definitions (3.41b), (3.34) and (3.35). Also, we'll need the following assumptions on the positions of the poles and zeroes of the function $\mathrm{e}^{\frac{\mathrm{i}}{2}\delta(\boldsymbol{w},p,s)}$, which were verified numerically for small $\mathrm{d} \leq 3$:

(i) The points where $\mathrm{e}^{\frac{\mathrm{i}}{2}\delta(\boldsymbol{w},p,s)}$ is singular do not coincide with the location of any zero of $\mathrm{e}^{\frac{\mathrm{i}}{2}\delta(\boldsymbol{w}',p,s)}$ with $\boldsymbol{w}'$ being some other solution set of (3.28).

(ii) All singularities of $\mathrm{e}^{\frac{\mathrm{i}}{2}\delta(\boldsymbol{w},p,s)}$ in the complex $s$ plane are simple poles. Notice that, in view of eq. (4.15), this implies that all of its zeroes are simple as well.

From assumption (i), any pole or zero of $\mathrm{e}^{\frac{\mathrm{i}}{2}\delta(\boldsymbol{w},p,s)}$ must appear as a pole/zero in the r.h.s. of (4.16). This way, one finds that the values of $s$ for which the first condition in (4.13) is obeyed are $s = \pm s_a$, with

$$s_a = \mathrm{i}\left(-p-\tfrac{1}{2}-a\right) = \mathrm{i}\left(\tfrac{n}{2}-\mathcal{S}-a\right) \qquad \text{and} \qquad 0 \leq a+\mathcal{S} < \tfrac{n}{2}, \quad a \in \mathbb{Z}, \tag{4.17}$$

where the bound on $a + \mathcal{S}$ comes from (4.14). Moreover, due to (ii) one can determine the number of solution sets $\boldsymbol{w}$ with (4.13) being satisfied at $s = s_a$ by reading off the multiplicity of that pole/zero from eqs. (4.16) and (4.6). This would coincide with the dimension of the linear space $\mathcal{H}_{p,s}^{(\mathrm{disc,d})}$, which is the span of all states of the form $|\psi_{p,s}^{(S^z)}(\boldsymbol{w})\rangle$ having fixed $\mathcal{S}$, $S^z$, $\mathrm{d}$ and $s$ with $\mathfrak{Im}(s) \neq 0$. One finds the number of such $\boldsymbol{w}$ to be $N_a^+(\mathrm{d})$.

Define the space $\mathcal{H}_{\mathcal{S},S^z}^{(\mathrm{disc})}$ as the linear span of all the states that appear in the scaling limit of $\mathcal{H}_{L|\mathcal{S},S^z}^{(\mathrm{disc})}$. These are isomorphic to $\mathcal{H}_{\mathcal{S},\mathcal{S}}^{(\mathrm{disc})}$ and the analysis of the quantization condition performed above implies that:

$$\mathcal{H}_{\mathcal{S},\mathcal{S}}^{(\mathrm{disc})} = \bigoplus_{s \in \Sigma^+ \cup \Sigma^-} \bigoplus_{\mathrm{d}=0}^{\infty} \mathcal{H}_{p,s}^{(\mathrm{disc,d})} . \tag{4.18}$$

Here $\Sigma^{\pm}$ denote the finite sets of pure imaginary numbers:

$$\Sigma^+ = \left\{s \,:\, \tfrac{n}{2}+\mathrm{i}s \in \mathbb{Z}, \ \mathfrak{Im}(s) \in (0,\tfrac{n}{2}]\right\}, \qquad \Sigma^- = \left\{s \,:\, \tfrac{n}{2}-\mathrm{i}s \in \mathbb{Z}, \ \mathfrak{Im}(s) \in (-\tfrac{n}{2},0)\right\}, \tag{4.19}$$

which incorporate the bound on the imaginary part of $s$ (4.14). Each component $\mathcal{H}_{p,s}^{(\mathrm{d})}$ is finite dimensional and

$$\dim\left(\mathcal{H}_{p,s}^{(\mathrm{disc,d})}\right) = N_a^+(\mathrm{d}|\mathcal{S}) \qquad \text{with} \qquad a = \tfrac{n}{2}-\mathcal{S} \pm \mathrm{i}s \in \mathbb{Z} \tag{4.20}$$

(here and below, when a condition involving $\pm\mathrm{i}s$ appears we mean it is to be satisfied for some choice of the sign $+$ or $-$).

The following comment is in order here. For the case $a < 0$, the integers $N_a^+(\mathrm{d}|\mathcal{S})$ (4.7) are all zero for $\mathrm{d} = 0$:

$$N_a^+(0|\mathcal{S}) = 0 \qquad \text{for} \qquad a = -1,-2,-3,\dots . \tag{4.21}$$

As a result, for the primary Bethe states the limiting values of $\mathfrak{Im}(b(L))$ are given by $s = \pm s_a$ (4.17) with the extra condition imposed that $a \geq 0$. Thus one recovers the results of the

work [23], see also formula (3.14). Of course, RG trajectories exist for the spin chain which are not primary Bethe states that are labeled by $s = \pm s_a$ with $a < 0$.

We conjecture that any RG trajectory of the lattice model with given $U_q\big(\mathfrak{sl}(2)\big)$ spin $\mathcal{S}$ and eigenvalue of the $z$-projection of the total spin operator $S^z$ belongs either to $\mathcal{H}^{(\mathrm{cont})}_{L|\mathcal{S},S^z}$ or $\mathcal{H}^{(\mathrm{disc})}_{L|\mathcal{S},S^z}$. Thus, the full space of low energy states of the lattice system in the scaling limit becomes the linear space

$$\mathcal{H} = \mathcal{H}^{(\mathrm{cont})} \oplus \mathcal{H}^{(\mathrm{disc})} \tag{4.22}$$

with

$$\mathcal{H}^{(\mathrm{cont})} = \bigoplus_{\mathcal{S}=0}^{\infty} \bigoplus_{S^z=-\mathcal{S}}^{\mathcal{S}} \mathcal{H}^{(\mathrm{cont})}_{\mathcal{S},S^z}, \qquad \mathcal{H}^{(\mathrm{disc})} = \bigoplus_{\mathcal{S}=0}^{\infty} \bigoplus_{S^z=-\mathcal{S}}^{\mathcal{S}} \mathcal{H}^{(\mathrm{disc})}_{\mathcal{S},S^z}. \tag{4.23}$$

We call $\mathcal{H}^{(\mathrm{cont})}$ the 'continuous spectrum' due to the presence of a direct integral over $s$ in its decomposition, see (4.11). The space $\mathcal{H}^{(\mathrm{disc})}$ will be referred to as the 'discrete spectrum'.

## 4.2 $\mathcal{W}_\infty$ algebra

In the scaling limit the critical lattice system possesses extended conformal symmetry. The corresponding algebra is expected to be the $\mathcal{W}_\infty$ algebra from ref. [46] with central charge $c < 2$. This is the same one that appears in the scaling limit of the staggered six-vertex model with (quasi-)periodic BCs [15, 26]. Among other things, such a statement implies that the graded linear spaces

$$\bigoplus_{\mathrm{d}=0}^{\infty} \mathcal{H}^{(\mathrm{d})}_{p,s}, \qquad \mathcal{H}^{(\mathrm{d})}_{p,s} = \begin{cases} \mathcal{H}^{(\mathrm{cont,d})}_{p,s} & \text{for} \quad s \in \mathbb{R} \\ \mathcal{H}^{(\mathrm{disc,d})}_{p,s} & \text{for} \quad p + \frac{1}{2} \pm is \in \mathbb{Z} \end{cases} \tag{4.24}$$

are isomorphic to a (irreducible) representation of $\mathcal{W}_\infty$. Then formulae (4.23), (4.11) and (4.18) would provide a classification of the space of states $\mathcal{H}$ occuring in the scaling limit of the lattice model in terms of irreps of the algebra of extended conformal symmetry. In order to demonstrate this we briefly mention some details concerning the $\mathcal{W}_\infty$ algebra and its representations, while referring the reader to section 16 of ref. [15] for a deeper discussion.

The $\mathcal{W}_\infty$ algebra is generated by a set of currents $W_j(u)$ of Lorentz spin $j = 2, 3, \ldots$. These satisfy an infinite system of Operator Product Expansions (OPEs). Its first few members can be chosen to be

$$W_2(u)\,W_2(0) = \frac{c}{2u^4} - \frac{2}{u^2}\,W_2(0) - \frac{1}{u}\,\partial W_2(0) + O(1)$$

$$W_2(u)\,W_3(0) = -\frac{3}{u^2}\,W_3(0) - \frac{1}{u}\,\partial W_3(0) + O(1) \tag{4.25}$$

$$W_3(u)\,W_3(0) = -\frac{c(c+7)(2c-1)}{9(c-2)u^6} + \frac{(c+7)(2c-1)}{3(c-2)u^4}\,\big(W_2(u) + W_2(0)\big) - \frac{1}{u^2}\,\big(W_4(u) + W_4(0)$$

$$+\ W_2^2(u) + W_2^2(0) + \frac{2c^2 + 22c - 25}{30(c-2)}\,\big(\partial^2 W_2(u) + \partial^2 W_2(0)\big)\big) + O(1)\,,$$

where in the last line $W_2^2$ is a composite field which coincides with the first regular term in the OPE $W_2(u)W_2(0)$. Notice that there is some ambiguity in the definition of $W_j$ for $j \geq 3$. Apart from the freedom in the overall multiplicative normalization, $W_j \mapsto C W_j$, it is possible to add to $W_j$ any differential polynomial of Lorentz spin $j$ involving the lower spin currents $W_k$ with $k < j$. Here, the $W_3$ current was fixed by the requirement that it be a primary field of spin three, so that its OPE with $W_2$ takes the form of the second line in formula (4.25). As

for $W_4$, one can not arrange for it to be a primary field by adding linear combinations of $W_2^2$, $\partial W_3$ and $\partial^2 W_2$. Defined such that it appears in the OPE of $W_3(u)W_3(0)$ as above, it turns out that $W_2(u)W_4(0)$ takes a simpler form,

$$W_2(u)\,W_4(0) = \frac{(c+10)(17c+2)}{15(c-2)\,u^4}\,W_2(0) - \frac{4}{u^2}\,W_4(0) - \frac{1}{u}\,\partial W_4(0) + O(1)\,, \tag{4.26}$$

where the singular terms $\propto u^{-6}$ and $\propto u^{-3}$ are absent.

For the study of the $\mathcal{W}_\infty$ algebra it is useful to know that it admits a realization in terms of two independent chiral Bose fields. We normalize them as

$$\partial\vartheta(u)\,\partial\vartheta(0) = -\frac{1}{2u^2} + O(1)\,, \qquad\qquad \partial\varphi(u)\,\partial\varphi(0) = -\frac{1}{2u^2} + O(1)\,, \tag{4.27}$$

while $\partial\varphi(u)\partial\vartheta(0) = O(1)$. One may check that as a consequence of the free field OPEs, the currents

$$W_2 \;=\; (\partial\vartheta)^2 + (\partial\varphi)^2 + \frac{\mathrm{i}}{\sqrt{n+2}}\,\partial^2\varphi \tag{4.28}$$

$$W_3 \;=\; \frac{6n+8}{3n+6}\,(\partial\vartheta)^3 + 2(\partial\varphi)^2\partial\vartheta + \mathrm{i}\sqrt{n+2}\,\partial^2\varphi\,\partial\vartheta - \frac{\mathrm{i}n}{\sqrt{n+2}}\,\partial\varphi\,\partial^2\vartheta + \frac{n}{6(n+2)}\,\partial^3\vartheta$$

obey the algebra (4.25). The parameter $n$ entering above is related to the central charge $c$ as

$$c = \frac{2(n-1)}{n+2} \tag{4.29}$$

so that if $n$ is real and positive, the central charge $c$ is less than two. Notice that, while an expression for $W_4$ in terms of $\partial\vartheta$ and $\partial\varphi$ has not been provided, it can be deduced from the OPEs (4.25) and the formula (4.28) for $W_2$ and $W_3$. One simply computes $W_3(u)W_3(0)$ with $W_3$ written in terms of free fields and compares the coefficient $\propto u^{-2}$ with the same coefficient appearing in the last two lines of eq. (4.25). It turns out that the higher spin currents always appear in the OPEs involving the lower spin ones. This way, starting from (4.28) and recursively computing OPEs, one can determine the realization of $W_j$ in terms of the free fields $\partial\varphi$ and $\partial\vartheta$ for any $j = 4,5,6,\dots$.

A stepping stone for the construction of highest weight irreducible representations of the $\mathcal{W}_\infty$ algebra is the Verma module. It is defined using the Fourier modes of $W_j(u)$, which we assume to be periodic functions of the variable $u \sim u + 2\pi$:

$$W_j = -\frac{c}{24}\,\delta_{j,2} + \sum_{m=-\infty}^{\infty} \widetilde{W}_j(m)\,\mathrm{e}^{-\mathrm{i}mu}\,. \tag{4.30}$$

Introduce the highest state, which is specified by the conditions:

$$\widetilde{W}_j(m)\,|\boldsymbol{\omega}\rangle = 0 \qquad (\forall m > 0)\,, \qquad \widetilde{W}_j(0)\,|\boldsymbol{\omega}\rangle = \omega_j\,|\boldsymbol{\omega}\rangle \tag{4.31}$$

with $j = 2,3$. The highest weight is given by $\boldsymbol{\omega} = (\omega_2, \omega_3)$, where the component $\omega_2$ is equal to the conformal dimension of the highest state, while $\omega_3$ is the eigenvalue of $\widetilde{W}_3(0)$, which commutes with $\widetilde{W}_2(0)$. The Verma module is spanned by the states that are obtained by acting with the 'creation modes' of the spin 2 and spin 3 currents on the highest state:

$$\widetilde{W}_2(-\ell_1)\dots\widetilde{W}_2(-\ell_m)\,\widetilde{W}_3(-\ell_1')\dots\widetilde{W}_3(-\ell_{m'}')|\boldsymbol{\omega}\rangle \tag{4.32}$$

with $\ell_j$, $\ell'_{j'} \geq 1$. It possesses a natural grading given by

$$\mathrm{d} = \sum_{j=1}^{m} \ell_j + \sum_{j=1}^{m'} \ell'_j \tag{4.33}$$

and the dimensions of its level subspace with fixed $\mathrm{d}$ is the number of bi-partitions of $\mathrm{d}$, i.e., $\mathrm{par}_2(\mathrm{d})$ (3.29). In what follows we will parameterize the highest weight for the Verma module $\mathcal{V}_{\rho,\nu}$ as

$$\omega_2 = \frac{\rho^2 - \frac{1}{4}}{n+2} + \frac{\nu^2}{n} \tag{4.34}$$

$$\omega_3 = \frac{2\nu}{\sqrt{n}} \left( \frac{\rho^2}{n+2} + \frac{(3n+4)\,\nu^2}{3n\,(n+2)} - \frac{2n+3}{12\,(n+2)} \right).$$

This is motivated by the free field realization (4.28). Supposing that the highest state is an eigenvector of the operators $\int \mathrm{d}u\, \partial\vartheta(u)$ and $\int \mathrm{d}u\, \partial\varphi(u)$ with eigenvalues $\frac{\nu}{\sqrt{n}}$ and $\frac{\rho}{\sqrt{n+2}}$, respectively, formula (4.34) follows from (4.28). The highest weight is an even function of $\rho$. As a result the spaces $\mathcal{V}_{\rho,\nu}$ and $\mathcal{V}_{-\rho,\nu}$ should be identified. In the parameterization (4.34), the conformal dimensions of a state in the Verma module at level $\mathrm{d}$ is such that

$$\widetilde{W}_2(0) - \frac{c}{24} = \frac{\rho^2}{n+2} + \frac{\nu^2}{n} - \frac{1}{12} + \mathrm{d}, \tag{4.35}$$

which should be compared with eq. (3.13).

For generic complex values of $\rho$ and $\nu$ the Verma module $\mathcal{V}_{\rho,\nu}$ is an irreducible representation of the $\mathcal{W}_\infty$ algebra. However, with $\rho$, $\nu$ obeying certain constraints, the Verma module contains null vectors – highest states occurring at non-zero levels. Then the highest weight irrep $\mathcal{W}_{\rho,\nu}$ can be obtained from $\mathcal{V}_{\rho,\nu}$ by factoring out all of the invariant subspace(s) generated by the null vector(s). In view of applications to the scaling limit of the lattice model of particular interest is when $\rho = \pm\frac{1}{2}\big(r - m(n+2)\big)$ with $r, m = 1, 2, \ldots$ . In this case a null vector occurs at level $\mathrm{d} = mr$ and the Verma module splits into the direct sum of two representations, which are irreducible for generic $n$ and $\nu$:

$$\mathcal{V}_{\rho,\nu} = \mathcal{W}_{\rho,\nu} \oplus \mathcal{W}_{\rho',\nu} \qquad \text{with} \qquad \begin{array}{ll} \rho = \frac{1}{2}(r - m(n+2)) & (n,\ \nu - \text{generic}) \\[4pt] \rho' = \frac{1}{2}(r + m(n+2)) & (r, m = 1, 2, \ldots) \end{array}. \tag{4.36}$$

The space $\mathcal{W}_{\rho',\nu}$ is isomorphic to the Verma module and the dimensions of its level subspaces is $\mathrm{par}_2(\mathrm{d})$, while for $\mathcal{W}_{\rho,\nu}$, the level subspaces are $\mathrm{par}_2(\mathrm{d}) - \mathrm{par}_2(\mathrm{d} - mr)$ dimensional. Consider again the components $\mathcal{H}_{p,s}^{(\mathrm{cont},\mathrm{d})}$, which appear in the decomposition of the continuous spectrum of the space of states occurring in the scaling limit of the spin chain. Taking into account formulae (4.12) and (3.31) it is clear that

$$\mathcal{W}_{p,s} \cong \bigoplus_{\mathrm{d}=0}^{\infty} \mathcal{H}_{p,s}^{(\mathrm{cont},\mathrm{d})} \qquad \big(2p = 2\mathcal{S} + 1 - (n+2),\ s - \text{real}\big). \tag{4.37}$$

To describe the discrete spectrum in terms of irreps of the $\mathcal{W}_\infty$ algebra, it is necessary to analyze the case when $\nu$ is such that $\rho + \frac{1}{2} \pm i\nu$ is an integer for some choice of the sign $\pm$. As explained in, e.g., section 16.2 of reference [15] the Verma module with $\rho + \frac{1}{2} + i\nu = -a_+ = 0, \pm 1, \pm 2, \ldots$ contains a null vector at level $|a_+ + \frac{1}{2}| + \frac{1}{2}$, while for $-\rho + \frac{1}{2} + i\nu = -a_- = 0, \pm 1, \pm 2, \ldots$ there is a

null vector at level $|a_- + \frac{1}{2}| + \frac{1}{2}$. Assuming $\rho$ is generic for now the character of the irreducible representation,

$$\mathrm{ch}_{\rho,\nu}(q) \equiv \mathrm{Tr}_{\mathcal{W}_{\rho,\nu}} \left[ q^{\widetilde{W}_2(0) - \frac{c}{24}} \right], \tag{4.38}$$

is given by [48] (see also [47, 49])

$$\mathrm{ch}_{\rho,\nu}(q) = q^{-\frac{1}{12} + \frac{\nu^2}{n} + \frac{\rho^2}{n+2}} \left( \prod_{m=1}^{\infty} \frac{1}{(1-q^m)^2} \right) \sum_{m=0}^{\infty} (-1)^m \, q^{m|a + \frac{1}{2}| + \frac{m^2}{2}} \qquad \begin{matrix} \rho + \frac{1}{2} \pm i\nu = -a \in \mathbb{Z} \\ n, \rho \;\; \text{generic} \end{matrix}. \tag{4.39}$$

If, in addition to $\nu$ being constrained as above, $\rho \to \frac{1}{2}(2\mathcal{S} - n - 1)$ then the irrep $\mathcal{W}_{\rho,\nu}$ further breaks up into two irreducible representations. One of them is generated by the null vector which appears at level $2\mathcal{S} + 1$ and has highest weights $(\rho', \nu)$ with $\rho' = \frac{1}{2}(2\mathcal{S} + n + 3)$. Its character is given by (4.39) with $\rho$ replaced by $\rho'$ and $a \mapsto a' = -2(\mathcal{S} + 1) - a$. Taking the difference $\mathrm{ch}_{\rho,\nu}(q) - \mathrm{ch}_{\rho',\nu}(q)$ with $\rho \to \frac{1}{2}(2\mathcal{S} - n - 1)$ and $\rho' \to \frac{1}{2}(2\mathcal{S} + n + 3)$ yields for the character of the irreducible representation $\mathcal{W}_{\rho,\nu}$ with

$$\rho + \frac{1}{2} \pm i\nu = -a \in \mathbb{Z} \qquad \text{and} \qquad 2\rho = 2\mathcal{S} - n - 1 \tag{4.40}$$

that

$$\mathrm{ch}_{\rho,\nu} = q^{-\frac{1}{12} + \frac{\nu^2}{n} + \frac{\rho^2}{n+2}} \left( \prod_{j=1}^{\infty} \frac{1}{(1-q^j)^2} \right) \sum_{m=0}^{\infty} (-1)^m \, q^{\frac{m^2}{2}} \left( q^{m|a+\frac{1}{2}|} - q^{2\mathcal{S}+1+m|2\mathcal{S}+a+\frac{3}{2}|} \right). \tag{4.41}$$

For the case $a \geq 0$ the above expression, apart from the overall factor $q^{-\frac{1}{12} + \frac{\nu^2}{n} + \frac{\rho^2}{n+2}}$, coincides with the generating function (4.7) for the integers $N_a^+(\mathrm{d}\,|\,\mathcal{S})$. This way, one concludes

$$\mathcal{W}_{p,s} \cong \bigoplus_{\mathrm{d}=0}^{\infty} \mathcal{H}_{p,s}^{(\mathrm{disc},\mathrm{d})} \qquad \left( p = \mathcal{S} + \frac{1}{2} - \frac{1}{2}(n+2), \; \frac{n}{2} - \mathcal{S} \pm is = a \in \mathbb{Z}_{\geq 0} \right). \tag{4.42}$$

The remaining case to be considered is when $-\mathcal{S} \leq a < 0$. The lower bound comes from the condition $s \in (-\frac{n}{2}, \frac{n}{2}]$ which implies that $\pm is = \frac{n}{2} - \mathcal{S} - a \leq \frac{n}{2}$. From the definition of the integers $N_a^+(\mathrm{d}\,|\,\mathcal{S})$ (4.7), which give the dimensions of the level subspaces $\mathcal{H}_{p,s}^{(\mathrm{disc},\mathrm{d})} \subset \mathcal{H}^{(\mathrm{disc})}$, one finds

$$\dim\left( \mathcal{H}_{p,s}^{(\mathrm{d})} \right) = 0 \qquad \text{for} \qquad \mathrm{d} = 0, 1, \ldots, |a| - 1 \qquad \left( -p - \frac{1}{2} \pm is = a \in \mathbb{Z}, \; -\mathcal{S} \leq a < 0 \right) \tag{4.43}$$

Thus the corresponding irrep (4.24) has highest state whose conformal dimension is given by:

$$\Delta = \frac{p^2}{n+2} + \frac{s^2}{n} + |a| \qquad \left( -\mathcal{S} \leq a < 0 \right). \tag{4.44}$$

This turns out to be an irreducible representation of the $\mathcal{W}_\infty$ algebra,

$$\mathcal{W}_{\rho,\nu} = \bigoplus_{\mathrm{d}=0}^{\infty} \mathcal{H}_{p,s}^{(\mathrm{d})}, \tag{4.45}$$

with highest weight parameterized as in (4.34), where

$$\rho = \mathcal{S} + \frac{1}{2}, \qquad \nu = \begin{cases} s - \frac{in}{2} & \text{for} \;\; (-is) > 0 \\ s + \frac{in}{2} & \text{for} \;\; (-is) < 0 \end{cases} \qquad \left( \frac{n}{2} - \mathcal{S} \pm is = a \in \mathbb{Z}_{<0}, \; -\mathcal{S} \leq a < 0 \right).$$

Assuming $n$ is irrational, the character of such a representation is given by

$$
\mathrm{ch}_{\rho,\nu}(\mathsf{q}) = \mathsf{q}^{-\frac{1}{12}+\frac{\nu^2}{n}+\frac{\rho^2}{n+2}} \left( \prod_{m=1}^{\infty} \frac{1}{(1-\mathsf{q}^m)^2} \right) \sum_{m=0}^{\infty} (-1)^m \, \mathsf{q}^{\frac{m^2}{2}} \left( \mathsf{q}^{m||\rho|-|\nu||} - \mathsf{q}^{(m+1)(|\rho|+|\nu|+1)-\frac{1}{2}} \right).
$$
(4.46)

One can check that the dimensions of the level subspaces, obtained by expanding $\mathrm{ch}_{\rho,\nu}(\mathsf{q})$ in a series in $\mathsf{q}$, coincides with the integers $N_a^+(\mathsf{d}|\mathcal{S})$ with $-\mathcal{S} \le a < 0$ and $\mathsf{d} = |a|, |a|+1, |a|+2, \dots$.

Finally, we mention that the states $|\psi_{p,s}^{(S^z)}(\boldsymbol{w})\rangle \in \mathcal{W}_{\rho,\nu}$ appearing in the scaling limit of the Bethe states, see eq.(4.9), have an important interpretation. They are the simultaneous eigenstates of the family of commuting operators known as the quantum AKNS integrable structure [50,51]. The function $\mathrm{e}^{\frac{\mathrm{i}}{2}\delta}$ (3.41b) entering into the quantization condition coincides with the eigenvalue of a certain so-called reflection operator [52] computed on $|\psi_{p,s}^{(S^z)}(\boldsymbol{w})\rangle$, see ref. [45] for details.

# 5 Partition function in the scaling limit

In the case of the lattice model with (quasi-)periodic Boundary Conditions (BCs) imposed, it was proposed in the work [16] and then verified numerically in ref. [15] that the partition function appearing in the scaling limit of the lattice system, $Z^{(\mathrm{scl})}$, coincides with twice the partition function for the 2D Euclidean black hole CFT. The latter was constructed in refs. [20,21] by computing a functional integral with the worldsheet being taken to be a torus. The results presented in the previous section allow one to easily compute $Z^{(\mathrm{scl})}$ for the staggered six-vertex model subject to $U_q\big(\mathfrak{sl}(2)\big)$ invariant open BCs. One may expect $\frac{1}{2} Z^{(\mathrm{scl})}$ to coincide with the partition function for the 2D Euclidean black hole CFT on the open segment $x \in (0, R)$, with certain conditions imposed on the fields at $x = 0, R$.

Consider the lattice partition function

$$
Z_L^{(\mathrm{lattice})} = \mathrm{Tr}_{\mathscr{V}_{2L}} \big[ \mathrm{e}^{-M\,\mathbb{H}} \big],
$$
(5.1)

where the Hamiltonian $\mathbb{H}$ is given by (1.3) with $q = \mathrm{e}^{\frac{\mathrm{i}\pi}{n+2}}$ and $n \ge 0$, while the trace is taken over the $2^{2L}$ dimensional space of states: $\mathscr{V}_{2L} = \mathbb{C}_1^2 \otimes \mathbb{C}_2^2 \otimes \dots \otimes \mathbb{C}_{2L}^2$. Keeping the ratio

$$
\tau = \frac{v_{\mathrm{F}} M}{2L}
$$
(5.2)

fixed as $L \to \infty$, one finds that the large $L$ behaviour of the lattice partition function is given by

$$
Z_L^{(\mathrm{lattice})} \asymp \mathrm{e}^{-MLe_\infty - Mf_\infty} \, Z^{(\mathrm{scl})}.
$$
(5.3)

Here $Z^{(\mathrm{scl})}$ takes the form of a trace over the space of states $\mathcal{H}$ appearing in the scaling limit of the lattice model:

$$
Z^{(\mathrm{scl})} = \mathrm{Tr}_{\mathcal{H}} \big( \mathsf{q}^{\hat{H}_{\mathrm{CFT}}} \big) \qquad \text{with} \qquad \mathsf{q} = \mathrm{e}^{-2\pi\tau}.
$$
(5.4)

It involves the 'CFT Hamiltonian' $\hat{H}_{\mathrm{CFT}}$ which when restricted to the finite dimensional spaces $\mathcal{H}_{p,s}^{(\mathrm{cont},\mathsf{d})}$ or $\mathcal{H}_{p,s}^{(\mathrm{disc},\mathsf{d})}$ appearing in the decomposition of $\mathcal{H}$ coincides with the identity operator multiplied by the factor

$$
E_{\mathrm{CFT}} = \frac{p^2}{n+2} + \frac{s^2}{n} - \frac{1}{12} + \mathsf{d}.
$$
(5.5)

Notice that the asymptotic formula for the energy (3.3) can be re-written as the formal relation

$$\hat{H}_{\text{CFT}} = \operatorname*{slim}_{L \to \infty} \frac{L}{\pi v_{\text{F}}} \left( \mathbb{H} - L e_{\infty} - f_{\infty} \right). \tag{5.6}$$

In subsection 4.1 the space of states $\mathcal{H}$ was expressed as a direct sum of the continuous spectrum $\mathcal{H}^{(\text{cont})}$ and the discrete one $\mathcal{H}^{(\text{disc})}$, see formula (4.22). The contribution of the states to the trace in eq. (5.4) for each of these spaces will be denoted as $Z^{(\text{cont})}$ and $Z^{(\text{disc})}$, respectively, so that

$$Z^{(\text{scl})} = Z^{(\text{cont})} + Z^{(\text{disc})}, \tag{5.7}$$

where

$$Z^{(\text{disc})} = \operatorname{Tr}_{\mathcal{H}^{(\text{disc})}} \left( \mathsf{q}^{\hat{H}_{\text{CFT}}} \right), \qquad Z^{(\text{cont})} = \operatorname{Tr}_{\mathcal{H}^{(\text{cont})}} \left( \mathsf{q}^{\hat{H}_{\text{CFT}}} \right). \tag{5.8}$$

Let's first focus on the computation of $Z^{(\text{disc})}$. The space $\mathcal{H}^{(\text{disc})}$ is made up of the components $\mathcal{H}_{\mathcal{S},\mathcal{S}^z}^{(\text{disc})} \cong \mathcal{H}_{\mathcal{S},\mathcal{S}}^{(\text{disc})}$, which admit the decomposition (4.18) into finite dimensional spaces. Introduce the notation:

$$\chi_{a,\mathcal{S}}(\mathsf{q}) = \mathsf{q}^{-\frac{(\frac{n}{2}-\mathcal{S}-a)^2}{n} + \frac{p^2}{n+2} - \frac{1}{12}} \left( \prod_{j=1}^{\infty} (1-\mathsf{q}^j)^{-2} \right) \sum_{m=0}^{\infty} (-1)^m \left( 1-\mathsf{q}^{(1+m)(2\mathcal{S}+1)} \right) \mathsf{q}^{ma+\frac{m(m+1)}{2}}, \tag{5.9}$$

where, aside from the prefactor, the function $\chi_{a,\mathcal{S}}(\mathsf{q})$ coincides with the generating function for the dimensions of the level subspaces $\mathcal{H}_{\mathcal{S},\mathcal{S}}^{(\text{disc,d})}$, see eqs. (4.20) and (4.7). Then, the contribution of the discrete spectrum to the partition function reads as:

$$Z^{(\text{disc})} = \sum_{\mathcal{S} \geq 0} (2\mathcal{S}+1) \left( \chi_{-\mathcal{S},\mathcal{S}}(\mathsf{q}) + 2 \sum_{\substack{a \in \mathbb{Z} \\ 0 < a+\mathcal{S} < \frac{n}{2}}} \chi_{a,\mathcal{S}}(\mathsf{q}) \right). \tag{5.10}$$

Each term in the sum over $\mathcal{S}$ has multiplicity $(2\mathcal{S}+1)$ as a result of the $U_q\big(\mathfrak{sl}(2)\big)$ symmetry of the lattice model. Also, for every state with given $s = s_a$ (4.17) there exists another one with $s = -s_a$ which yields the same contribution to the partition function, except for the case when $s = \pm \frac{\text{i}n}{2}$, where they are identified as the same state. This explains why the functions $\chi_{a,\mathcal{S}}(\mathsf{q})$ come with a factor of two except the one with $a = -\mathcal{S}$ (recall that $s_a = \text{i}(\frac{n}{2}-\mathcal{S}-a)$ and hence $s_a = \frac{\text{i}n}{2}$ for $a = -\mathcal{S}$).

The contribution of the continuous spectrum to the partition function is given by

$$Z^{(\text{cont})} = \sum_{\mathcal{S} \geq 0} (2\mathcal{S}+1) \int_{-\infty}^{\infty} ds \sum_{\text{d} \geq 0} \rho_p^{(\text{d})}(s) \, \mathsf{q}^{\frac{s^2}{n} + \frac{p^2}{n+2} - \frac{1}{12} + \text{d}}. \tag{5.11}$$

Here $\rho_p^{(\text{d})}(s)$ is the density of states defined in formulae (4.5) and (4.6), while recall that $2p = 2\mathcal{S}+1-(n+2)$. Notice that $Z^{(\text{cont})}$ becomes singular as $L \to \infty$:

$$Z^{(\text{cont})} = Z^{(\text{sing})} + O(1), \tag{5.12}$$

where the singular part goes as $\log(L)$ and reads as

$$Z^{(\text{sing})} = \sqrt{\frac{n}{2\tau}} \frac{\log\big(2^{\frac{n+2}{n}} L/L_0\big)}{\pi \mathsf{q}^{\frac{1}{24}} \prod_{m=1}^{\infty} (1-\mathsf{q}^m)} \sum_{\mathcal{S}=0}^{\infty} (2\mathcal{S}+1) \, \mathsf{q}^{-\frac{1}{24} + \frac{p^2}{n+2}} \frac{1-\mathsf{q}^{2\mathcal{S}+1}}{\prod_{m=1}^{\infty} (1-\mathsf{q}^m)}. \tag{5.13}$$

The factor out the front of the sum is easily recognized to be the partition function of a boson taking values in the segment $\sim \log(L)$ with Neumann BCs imposed at the endpoints of the field at $x = 0, R$. As for the remaining term,

$$\sum_{\mathcal{S}=0}^{\infty} (2\mathcal{S}+1) \, \mathsf{q}^{-\frac{1}{24} + \frac{p^2}{n+2}} \frac{1-\mathsf{q}^{2\mathcal{S}+1}}{\prod_{m=1}^{\infty} (1-\mathsf{q}^m)}, \tag{5.14}$$

in all likelihood it corresponds to a boundary state which is a superposition of Ishibashi states associated with a degenerate representation of the Virasoro algebra with generic central charge $c$ (see ref. [53] for the $c = 1$ case). Note that (5.14) also appears in the scaling limit of the $XXZ$ spin $\frac{1}{2}$ chain with open $U_q\big(\mathfrak{sl}(2)\big)$ invariant BCs imposed [54].

Formulae (5.10) and (5.11) do not seem to correspond to the published results in the literature concerning branes in the 2D Euclidean black hole CFT, in particular, ref. [24]. As such, a separate investigation is required in order to establish a possible relation between the partition function $Z^{(\mathrm{scl})} = Z^{(\mathrm{cont})} + Z^{(\mathrm{disc})}$ and that of the black hole CFTs in the presence of boundaries.

## 6    Discussion

In this work the universal behaviour of the staggered six-vertex model with $U_q\big(\mathfrak{sl}(2)\big)$ invariant boundary conditions imposed was considered. We focused on the so-called self-dual case in the critical regime (1.2). The problem was reformulated in the Hamiltonian picture, where a central rôle belongs to $\mathbb{H}$ (1.3), which commutes with the transfer-matrix of the vertex model. The study of the $1/L$ corrections to the $L \to \infty$ behaviour of the energy of $\mathbb{H}$ for the low energy states allowed us to extract the spectrum of scaling dimensions of the statistical system. Our treatment, involving the use of novel numerical and analytic techniques, represents an advance on the type of analysis of the scaling limit of integrable lattice systems with open boundary conditions that typically exists in the literature.

The numerical construction of the RG trajectories was achieved via the method of the $Q$ operator. A key rôle was played by the formula (2.12) for the matrix elements of $\mathbb{Q}(\zeta)$, valid for a one parameter family of open boundary conditions in the sector $S^z = 0$. This is an original result of our work. The advantage of (2.12), as opposed to the expressions for $\mathbb{Q}(\zeta)$ appearing in the literature [28–31], is that it contains no infinite sums; works literally for any (generic) complex values of $q$ and the boundary parameter $\epsilon$; and can be programmed efficiently on the computer. We believe that, in view of the important applications of the $Q$ operator, it may be worthwhile to extend (2.12) to the other sectors of the Hilbert space with $S^z \neq 0$ and the case of more general open boundary conditions.

The powerful analytic technique, which was crucial to our investigation, is the ODE/IQFT approach to the study of the scaling limit of integrable, critical lattice systems. For the staggered six-vertex model in the regime (1.2) with quasi-periodic boundary conditions imposed, it was developed in the works [15, 26]. We found that it was applicable to the case of $U_q\big(\mathfrak{sl}(2)\big)$ invariant open boundary conditions as well. This points to the versatility of the ODE/IQFT approach, where the analysis for one set of boundary conditions can be readily carried over to another.

As was already observed in refs. [22, 23], the set of scaling dimensions of the statistical system possesses a continuous component, labeled by the quantum number $s = \mathrm{slim}_{L\to\infty} b(L) \in \mathbb{R}$ with $b(L)$ from eq. (3.6) [23]. One of the results of this work is the explicit formula for the density of states $\rho^{(\mathrm{d})}(s)$ (4.5), (4.6), which characterizes the continuous spectrum. In addition, we studied the RG trajectories $\{|\Psi_L\rangle\}$, where $s$ becomes a pure imaginary number in the scaling limit. Building on the analysis of [23], the discrete set $\Sigma \equiv \Sigma^+ \cup \Sigma^-$ (4.19) of all admissible values of pure imaginary $s$ was found. We also determined the dimension of the linear span of states occuring in the scaling limit of $|\Psi_L\rangle$ with given $s \in \Sigma$ and conformal dimensions $\Delta$ (3.13).

Our work includes a full characterization of the linear space $\mathcal{H}$ appearing in the scaling

limit of the space of low energy states of the lattice system. To describe it, $\mathcal{H}$ was decomposed into a direct sum of the 'continuous spectrum' $\mathcal{H}^{(\text{cont})}$ and the 'discrete spectrum' $\mathcal{H}^{(\text{disc})}$. The former, when expressed in terms of finite dimensional spaces, involves a direct integral over $s$, see eqs. (4.23), (4.10), (4.11), while the latter contains a direct sum (4.18). We explained how the graded linear spaces $\oplus_{\mathrm{d}=0}^{\infty} \mathcal{H}_{p,s}^{(\text{cont,d})}$ and $\oplus_{\mathrm{d}=0}^{\infty} \mathcal{H}_{p,s}^{(\text{disc,d})}$ are irreps. of the $\mathcal{W}_{\infty}$ algebra – the algebra of extended conformal symmetry of the model.

Perhaps the most interesting question is the relation between the scaling limit of the lattice system and the 2D Euclidean/Lorentzian black hole CFTs [17–19]. We believe that the formula for the partition function $Z^{(\text{scl})}$ provided in sec. 5 may be of help. Unfortunately, it does not seem to correspond to known results in the literature on branes in the 2D black hole CFTs. It is likely that progress in this direction would require a separate and detailed study.

## Acknowledgments

The authors are grateful to M. Flohr and V. Schomerus for their interest in the work and for valuable discussions. In addition, we would like to thank S. Ribault for including an enlightening discussion about the CFT interpretation of our results in his referee report. GK would also like to thank V. V. Bazhanov, S. L. Lukyanov and J. Teschner for useful and stimulating interactions.

The research of the authors is supported by the Deutsche Forschungsgemeinschaft (DFG) under grant No. Fr 737/9-2. Part of the numerical work has been performed on the LUH computer cluster, which is funded by the Leibniz Universität Hannover, the Lower Saxony Ministry of Science and Culture and the DFG.

## A   The asymptotic coefficients $\mathfrak{C}_{p,s}^{(\pm)}$

Here we provide a closed form expression for the coefficients $\mathfrak{C}_{p,s}^{(\pm)}(\boldsymbol{w})$ that were obtained in the work [45]. Among other things, they enter into the quantization condition (3.41).

One has

$$\mathfrak{C}_{p,s}^{(\pm)}(\boldsymbol{w}) = \mathfrak{C}_{p,s}^{(0,\pm)}\, \check{\mathfrak{c}}_{p,s}^{(\pm)}(\boldsymbol{w})\,, \tag{A.1}$$

where

$$\mathfrak{C}_{p,s}^{(0,\pm)} = \sqrt{\frac{2\pi}{n+2}}\;\; 2^{-p \pm \frac{\mathrm{i}(n+2)s}{n}}\,(n+2)^{-\frac{2p}{n+2}}\,\frac{\Gamma(1+2p)}{\Gamma(1+\frac{2p}{n+2})\,\Gamma(\frac{1}{2}+p \pm \mathrm{i}s)}\,, \tag{A.2}$$

while $\check{\mathfrak{c}}^{(\pm)}$ are one for $\mathrm{d}=0$. In the general case, they are given by the determinant of a $\mathrm{d} \times \mathrm{d}$ matrix:

$$\check{\mathfrak{c}}_{p,s}^{(\pm)}(\boldsymbol{w}) = \frac{(\mp 1)^{\mathrm{d}}\,\det\left(w_a^{b-1}\,U_a^{(\pm)}(b)\right)}{\prod_{a=1}^{\mathrm{d}} w_a\,\prod_{b>a}(w_b - w_a)\,\prod_{a=1}^{\mathrm{d}}\left(2p + 2a - 1 \pm 2\mathrm{i}s\right)} \tag{A.3}$$

with

$$
\begin{aligned}
U_a^{(\pm)}(D) &= (D-1)^2 - \left(2p + 2 + n \mp 2w_a + \sum_{b \neq a} \frac{4w_a}{w_a - w_b}\right)(D-1) \\
&\quad + \tfrac{1}{2}\,n^2 + \left(p + \tfrac{3}{2}\right)n \mp (n+1+2p+2\mathrm{i}s)\,w_a + 2p + 1 \\
&\quad + \left(\sum_{b \neq a} \frac{2w_a}{w_a - w_b}\right)^2 + \left(2\,(2p+1+n \mp 2w_a) - n\right)\sum_{b \neq a} \frac{w_a}{w_a - w_b}\,.
\end{aligned}
\tag{A.4}
$$

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
