# Peer review of "Scaling limit of the staggered six-vertex model with $U_q\big(\mathfrak{sl}(2)\big)$ invariant boundary conditions"

_SciPost Physics_

## Round 2 · Referee Report · Sylvain Ribault (Referee 1) · 2024-2-15

Report

The study of integrable statistical systems on a lattice is particularly interesting when it leads to a comparison with universal results from conformal field theory. This comparison becomes possible when the lattice size becomes large. But integrable systems' spaces of states tend to grow very fast with lattice size, so it is challenging to take the large size limit.

This article focuses on the staggered six-vertex model with a particular choice of boundary conditions, and manages to compute the partition function all the way to the infinite size limit. This yields exact results that can in principle be compared to CFT. I do not know integrable models well enough to judge how much of a technical feat this is, or how much confidence we should have in the derivation, which is largely conjectural. In any case, the results look very plausible from the point of view of CFT, and it would be very interesting to rederive them by CFT methods.

Unfortunately, there are no existing relevant CFT results, as the authors correctly point out. Let me first discuss this point in more detail. The staggered six-vertex model comes with an anisotropy parameter $q$ (1.2), which is then rewritten in terms of a positive real number $n$ (3.2). This is related to the central charge of the CFT by $c=\frac{2(n-1)}{n+2}$ (4.29). In existing CFT works such as Ref. [24], the CFT is described as a gauged WZW model $SL(2)/U(1)$; in Section 4.2 the authors use an equivalent formulation in terms of a $W_\infty$ algebra. The gauged WZW model comes with a level $k=-n$, in terms of which the central charge reads $c= \frac{3k}{k-2}-1$. The problem is that the CFT results are valid for $k>2$, whereas we would need $k<0$.

Could we just analytically continue the CFT results? This has been done in [15] for the torus partition function, leading to a non-trivial agreement between the lattice model and the CFT. However, there is no guarantee that this should work for cylinder partition functions. In fact, on the CFT side, the issue of analytic continuation has not been directly addressed in the $SL(2)/U(1)$ model. A very similar situation arises in Liouville theory, which also has a continuous spectrum. (Liouville theory is not just an analog of the $SL(2)/U(1)$ model, there is also a direct relation, see hep-th/0502048). The similar problem would be to analytically continue Liouville theory from $c>25$ to $c<1$. While this works well for the torus partition function, this fails for other correlation functions: more precisely, the analytic continuation is possible for any $c\in\mathbb{C}$, except precisely the half-line $c<1$, see for instance hep-th/0505063. This problem with analytic continuation is expected to affect the boundary CFT as well. As a result, boundary Liouville theory is not well-understood for $c<1$, see the discussion in 2309.10846. It is not even clear whether there are such objects as ZZ branes and FZZT branes for $c<1$. Even less is known on the boundary $SL(2)/U(1)$ model with $k<0$.

To make matters worse, spectral densities such as 24 depend on the boundary reflection coefficients. But reflection coefficients are not invariant under field renormalizations, so they are not truly universal quantities, and any discrepancy could be attributed to a different choice of normalization.

Therefore, even if the results of [24] were complete and reliable, they would not be applicable. From the CFT point of view, the present article provides an interesting challenge together with very precise hints. However, the priority would probably be to first understand boundary Liouville theory with $c<1$, which is more fundamental and presumably easier. This raises the issue of finding a lattice integrable model whose limit would be Liouville theory, rather than the more complicated $SL(2)/U(1)$ model: apparently such a model is not known. (Notwithstanding the results of 1509.03538, conformal loop ensembles are not related to Liouville theory, in particular their spectrum is discrete.)

Nevertheless, a potentially relevant piece of information could be salvaged from [24]: the boundary conditions that lead to partition functions with continuous terms come with continuous boundary parameters. This is a robust prediction of boundary CFT that I would expect to be applicable to the limit of the staggered six-vertex model. However, the present article's partition function (5.10), (5.11) does not come with any boundary parameter. The CFT therefore suggests that this partition function could be deformed, without breaking the $W_\infty$ symmetry. Does there exist a corresponding deformation of the staggered six-vertex model, and would it preserve the $U_q(sl(2))$ symmetry?

The above discussion goes beyond the scope of the present article, but it would be very interesting if the authors had comments on some of these points. Moreover, the article could be improved by discussing the values of the central charge in more detail. In addition, I have a few questions, and suggestions for improving clarity:

  1. A table of contents would be welcome.

  2. When diagonalizing the Hamiltonian, why can we restrict to the zero-eigenspace of the total spin operator (1.4)? Is it because other eigenspaces are deduced by acting with $U_q(sl(2))$ operators?

  3. On page 5, does the first paragraph mean that in order to follow the trajectory $L\to L+2$, we should look at Bethe roots rather than energies? This paragraph is either too long or too short: if this aspect is not omitted, it would be good to explain it in more detail.

  4. The meaning of the scaling limit, and of the symbol 'slim' that appears in (3.11), could be explained more precisely. Apparently, we do not want to follow a particular sequence of values of $b(L)$, but rather the set of these numbers? Can we define the resulting density $\rho(s)$ from the set of values of $b(L)$? (For an example of a precise definition of a continuous CFT spectrum as a limit of discrete spectrums, see hep-th/0107118.)

  5. Page 12, the last paragraph of Section 3.1 seems disconnected from the rest, and at the first reading it is not clear if it is relevant. Later, in Sections 3.3 and 4.1, it becomes apparent that this is the origin of the discrete part of the spectrum. It would be nice to comment on this issue in more detail. In particular, are the values of $s$ in (3.14) the only possibilities? Or at least, are the corresponding RG trajectories distinguished in some sense? How much arbitrariness is there in the construction of RG trajectories and determination of their scaling limits?

  6. The appearance of a new parameter $\mu$ in (3.15) is hard to swallow. Apparently this parameter is related to $\zeta$, with a relation that is implicitly written in (3.21). It might be helpful to explain more clearly the origin of $\mu$.

  7. Eq. (3.30) smells strongly of null vectors, and indeed it seems that it is related to the null vectors of Section 4.2. This relation between analytic properties of differential equations, and the algebraic construction of null vectors of W-algebras, is interesting and non-trivial. How new is it? It would be nice to state it more explicitly, if only as a conjecture, and to comment on its origin.

  8. Below (3.42), does "the above relation" refer to (3.40)?

  9. The crucial conjecture on page 18 would deserve more explanations, as its statement is very technical. Do I understand correctly that $b_*(L)$ is an approximation of $b(L)$, that $b_*(L)$ can be computed from the quantization condition whereas $b(L)$ requires solving the Bethe ansatz equations, and that $b(L)$ and $b_*(L)$ are equivalent in some sense in the scaling limit? If $b_*(L)$ is much easier to compute than $b(L)$, could this be stated quantitatively, for example by comparing the values of $L$ that can be reached?

  10. For the sake of the comparison with CFT, it would be nice to know which symmetries the CFT boundary conditions should preserve. Since the partition function is a combination of $W_\infty$ characters, $W_\infty$ symmetry should be preserved. However, doesn't the $W_\infty$ algebra have an automorphism $W_k \to (-1)^kW_k$ i.e. $\vartheta \to -\vartheta$? In this case, we could have either untwisted boundary conditions $W_k = \bar W_k$ or twisted boundary conditions $W_k = (-1)^k \bar W_k$. (These would be A-branes and B-branes in the language of hep-th/0408172.) Do we know which case is relevant to the present article? Is there such a distinction for boundary conditions of the spin chain?

I am grateful to the authors for helpful correspondence.

---

## Round 2 · Referee Report · Anonymous (Referee 3) · 2024-3-14

Strengths

1-Tackles interesting problem, attempting to extend the approach of [15] to the $U_q(sl_2) $ invariant open staggered chaing.

2-Contains a surprising and novel construction of the $(S_z=0) $ Q-operator for this model.

3-Carries out a detailed analysis of the RG flows of Bethe roots in order to characterise the scaling behaviour.

4-Claims to find a negative results regarding the connection with 2D black holes in this model.

Weaknesses

1- The presentation style is somethimes confusing and lacking in enough detail to be able to reconstruct claimed results.

2- The novel results about the Q-operator are not explained.

Report

This paper examines the scaling limit of the staggered 6 vertex model, taking the general approach of [15] and other associated works and applying it to the case of an open system with $U_q(sl_2)$ invariant boundary conditions. This is an obvious case to examine, in particular as the $U_q(sl_2)$ invariance simplfies the analysis of Bethe states.

The paper is generally well-written and addresses a problem of current interest to the quantum intebrable and CFT communities. The analysis is carried out and with care and with some exceptions explained coherently. The key tools used are in some ways now standard: numerical solution of the Bethe Equations and the ODE/IM correspondence; but they are deployed in a creative way to this novel example.

One of the motivations for the current work was to extend the approach of [17-19] to understand the connection between the scaling limit and 2D black holes in the case of the U_q(sl_2) invariant boundary conditions. The authors have found no such connection in this case, and while this is a negative result it is still of significant interest.

One byproduct of the current work is a novel construction of a Q-opertator for this open system, and this construction is highlighted as one of the key results of the paper. The presentation of this section seems the weakest part of the paper to me, and it is in this part that I would request the following changes if the paper is to be accepted.

1) The Q-operator construction presented is surprising as it does not involve an infinite sum corresponding to the trace over an infinite-dimensional auxiliary space. Such a trace is present in all existing constructions, including [28], of closed and open Q-operators that follow the key work of Bazhanov, Lukyanov and Zamolodchikov. The current paper says the formula for Q is 'obtained based on the results of [28] ...' . It then goes on to say that 'some analysis was required in order to bring the expression [of [28]] to a form which is literally applicable to the case at hand'.

The formula (2.13) for the Q-operator (for the S_z=0 sector) which is indeed very different to [28] is then presented with no further explanation.

Given that formula (2.13) is remarkably simple, and surprising, I would request that details of the analysis involved be included in this paper.

2) The Q-operator given by (2.15) is presumably constructed to be the object that satisfies (2.19) and (2.1).

First of all, there is no proof that (2.15) satisfies (2.19) and (2.1). If it exists I would ask the authors to include such a proof as it is key to the understanding of the claimed form (2.15).

Secondly, the existence of such a Q satisfying (2.19), (2.10) and (2.18) and polynomiality implies the Bethe Equations (2.22). These Bethe Equations(2.22) also come from the orginal open algebraic Bethe ansatz of Skylanin as the authors point out. It is these Bethe equations that are used in the rest of the paper. What I am therefore failing to understand is why the Q-operator (2.15) is constructed and how it relates to the rest of the paper. Is the form (2.15) used anywhere in the paper? I am quite happy that this Q-operator construction is given as an interesting aside, but the authors state 'Below we present the explicit formula for the Q-operator [...] It allow us to go beyond the results of the previous papers [22,23]'. The authors should please clarify this point.

Finally it would be useful to have some comment about whether Q of (2.15) is the unique solution (presumably up to multiplicative factors) of (2.19) and (2.1) or not.

Requested changes

See main report

---

## Round 2 · Referee Report · Anonymous (Referee 2) · 2024-3-14

Report

This manuscript studies an integrable lattice model with non-compact degrees of freedom in the scaling limit, namely the staggered six-vertex model. More sepcifically, it is concerned with the case of open boundary conditions which respect the $U_q(sl(2))$ quantum group symmetry.

The manuscript presents the following advances: - The derivation of a closed form expression for the Baxter Q-operator of the six-vertex model for a certain class of boundary conditions and in a given symmetry sector. - The numerical study of Bethe root patterns, and the corresponding eigenvalues of the Hamiltonian $H$ and quasi-shift operator $B$ - The study of the asymptotic behaviour of Bethe roots and eigenvalues of $H,B$ in the scaling limit, through the ODE/IM correspondence - The interpretation of this scaling limit in terms of representations of the $W_\infty$ algebra, including characters and annulus partition function.

The construction of the staggered six-vertex model with $U_q(sl(2))$ boundary conditions was done in ref [22], and its finite-size spectrum was first studied in ref [23] by numerical solution of the Bethe Ansatz equations, and by the linear root density approach. The present manuscript adapts the approach developed for the periodic case in refs [15,26], to provide a deeper study of the model with $U_q(sl(2))$ boundary conditions. Moreover, it makes use of recent discoveries in quantum integrable models, to extract a tractable expression for the Q-operator, to be used for numerical study.

The manuscript presents substantial advances to the study of quantum integrable models, and it is well written. Therefore I recommend its publication in SciPost Physics, provided the authors address the following questions:

  1. If I understand correctly, the central result of Section 2 is the expression (2.15) for the Q-operator, derived from the results of ref [28]. This derivation should be explained in more detail.

  2. A well-known technique to find the Bethe root patterns from exact diagonalisation of small systems, is to compute numerically the coefficients of the eigenvalue T as a polynomial in the spectral parameter, and extract the coefficients of the Q polynomial from the TQ equation (2.19) [G. Albertini, S. Dasmahapatra, B.M. McCoy, Spectrum and Completeness of the Integrable 3-state Potts Model: A Finite Size Study, Int. J. Mod. Phys. A7, 1 (1992)]. This does not require an explicit expression for the Q-operator. I understand that the determination of (2.15) and (2.26) for the Q-operator has an interest in its own right, but could the authors discuss the compared advantages of their numerical scheme and of the more traditional one described above ?

  3. In Sections 3.2-3.3 the ODE/IQFT correspondence is given, in the case of $U_q(sl(2))$ boundary conditions. Can the authors include a discussion of the differences of this correspondence with respect to the periodic case ?

  4. In BCFT, when conformal boundary conditions are imposed at the ends of the 1d system, Cardy's scheme allows to predict, on the basis of fusion rules, which representations of the spectrum generating algebra (in the present case, $W_\infty$) contribute to the Hilbert space. Can the authors discuss if this approach is applicable here, even though the CFT is not rational ? More generally, are the representations (4.37) and (4.42) in correspondence with the ones contributing to the bulk CFT (i.e. the ones for the periodic Hamiltonian) ?

---

## Round 3 · Referee Report · Sylvain Ribault · 2024-4-5

Report

The authors have made a number of minor changes, which contribute to making the article clearer. I am happy that they found some of my suggestions helpful. I have no further changes to suggest.

---

## Round 3 · Referee Report · Anonymous · 2024-4-15

Report

I am happy that the authors have now addressed the two key concerns raised in my report on the earlier version of this paper. I am happy to recommend publication.

Recommendation

Publish (easily meets expectations and criteria for this Journal; among top 50%)

---

## Round 3 · Referee Report · Anonymous · 2024-5-13

Report

The authors have addressed in a satisfactory way the points raised in my previous report.

Recommendation

Publish (easily meets expectations and criteria for this Journal; among top 50%)

---

## Round 3 · Author Response

We would like to thank the referees for a careful reading of our paper and for the useful comments. Below is our reply to the issues they've raised.

The second and third referee reports give some criticisms of our discussion of the $Q$ operator in section two. Among other things, they recommend adding to the manuscript a short explanation of how the formula for the $Q$ operator (2.15) is related to the results of [28]. We have done this in the revised version, see the paragraph below eq. (2.18). It is clear from the updated paper that the relation is not particularly direct. As such, although we have checked for small lattice sizes $N=2,3,4,...$ that the commutativity condition (2.1) and $TQ$ relation (2.19) are satisfied, their proof for general $N$ as well as questions concerning uniqueness can not be literally carried over from the results of ref. [28]. We believe that such issues are beyond the scope of our study and should be addressed in a separate paper. Also, it is explained in the opening paragraphs of section 2 why the $Q$ operator is important for the construction of RG trajectories, namely, that it allows one to extract the Bethe roots for a low energy state at intermediate values of $N\sim 20$. In the 2nd referee report another approach is mentioned for finding the Bethe roots corresponding to a given state which is usually referred to as the McCoy method. It starts by computing the eigenvalues of the transfer-matrix and then recovering the eigenvalue of the $Q$ operator from the $TQ$ relation. When a numerically efficient representation for $\mathbb{Q}(\zeta)$ exists, we find that the $Q$ operator approach is more efficient than the McCoy method. In our experience, the latter leads to greater numerical errors which, in practice, limits its applicability to smaller values of intermediate $N$.

The first referee report contains a very interesting discussion regarding the problem of the identification of the boundary CFT underlying the scaling limit of the lattice system. We are grateful for this expert's perspective on the results of our study, which identifies many fruitful areas for future research. At the same time, we would prefer not to comment much on the CFT interpretation of our results in the paper. The reason is that, at most, we can only give a speculative discussion, which we would prefer to avoid in case it turns out later that we say something incorrect. The questions being raised require a separate, detailed investigation that could form the subject matter of another work. In the updated manuscript we have added an acknowledgement to Sylvain Ribault for his valuable scientific input.

Some comments to the remaining minor issues raised by the referees are:

Referee report 2: 3. The differences in the ODE/IQFT correspondence between the case with periodic Boundary Conditions (BCs) and open $U_q(\mathfrak{sl}(2))$ invariant BCs is what one would expect from the basic principles of boundary CFT. Namely, instead of there being two ODEs appearing in the scaling limit of a low energy Bethe state --- one for each chirality --- there is only one differential equation for open BCs. Also, the precise relation between the invariants labeling the RG trajectories and those entering into the ODE are different. The interested reader can compare what is written in section 3.2 in the manuscript with the discussion in, e.g., section 11 of ref.[15] for details.

  1. Cardy's scheme was developed in the context of the minimal models. Its application to the CFT underlying the critical behaviour of the staggered six-vertex model, which is non-rational, would require a significant extension. This is well beyond the scope of our paper. Also, the low energy states in the scaling limit organize into different irreps of the ${\cal W}_\infty$ algebra in the case of periodic BCs as opposed to open, $U_q(\mathfrak{sl}(2))$ invariant BCs.

Referee report 1: 1. This is a good idea and we have added a contents page in the revised manuscript.

  1. What is being said is correct -- all other eigenstates can be obtained from those in the sector $S^z=0$ via the $U_q(\mathfrak{sl}(2))$ raising and lowering operators. We make comments along these lines, see, e.g., the two sentences below eq.\,(2.12).

  2. Yes indeed, RG trajectories are constructed via an analysis of the Bethe Ansatz equations --- as is usual in the finite size analysis of Yang-Baxter integrable lattice models. We believe that our two paragraph discussion is sufficient, and we refer the reader to the works [15] and [26] if extra clarification is required. Also, we can not remove the paragraphs entirely as is suggested. They are needed to motivate the importance of the $Q$-operator, without which an analysis of the type performed in our work would be considerably more difficult to carry out. We have tried our best to keep a balance here.

  3. We agree that in order to define the density $\rho(s)$, it is enough to consider the set of values of $b(L)$ at large $L$. However, the paragraph containing formula (3.11) that involves the symbol `${\rm slim}$' concerns a different problem, namely, how to suitably define the scaling limit of an individual low energy state of the lattice model. This turns out to be rather subtle, which motivates the use of a separate symbol for the scaling limit. An additional clarifying comment has been included in the revised manuscript.

  4. Section 3.1 is mainly devoted to a review of the works [22,23], where the lattice model was also studied. Formula (3.14) is an important result of ref. [23], which is why that paragraph was included. Later, in section 4.1, we give all the possible values of pure imaginary $s$ that can occur in the scaling limit of a low energy Bethe state (see (4.17)). We discuss the relation of our results to those of [23] in the paragraph containing formula (4.21).

  5. We agree that the appearance of the parameter $\mu$ entering into (3.15) may be confusing and that its relation to $\zeta$ was only implicitly given later. In the revised version of the manuscript we give this relation immediately after (3.15).

  6. The r.h.s. of formula (3.30) coincides with the dimension of the level subspace of an irreducible module of the $W_\infty$ algebra, which is obtained from the Verma module by removing the descendents of the null vector that occurs at level ${\tt d}-2{\cal S}-1$. This is discussed further around formulae (4.36) and (4.37) in section 4. That the algebraic system (3.27), which comes from an analysis of the ODE, correctly accounts for the case when the Verma module develops null vectors has been observed before. A detailed discussion is contained in, e.g., section 12 of the work [15].

  7. By ``the above relation'' we meant formulae (3.40)-(3.42) in the original submission, which should be treated as one set. To make this clear we have changed the numbering to (3.41a)-(3.41c) and refer to this set as (3.41) later in the revised version.

  8. The fact that we defined $b(L)$ to take values in the strip $-\frac{n}{2}<\Im m\big(b(L)\big)\le\frac{n}{2}$ (see eq. (3.8)) is the source of the additional complications in the conjecture on page 18. They are purely of a technical nature, included to ensure the formal correctness of the statement. It is right to say that $b_(L)$ coming from the quantization condition approaches $b(L)$ computed from the Bethe Ansatz equations in the scaling limit and that there is essentially a one-to-one relation between the sets of values ${b_(L)}$ and ${b(L)}$ at $L\gg1$. We believe that fig. 6 gives a good demonstration of this. In principle, for a given RG trajectory, the numerical computation of $b(L)$ is possible for any $L\gg 1$. However, obtaining $b_*(L)$ from the quantization condition requires much less programming and computing resources. The quantization condition can be analysed analytically as well, which is its main advantage. We have added a few sentences in the preamble of section 4.1 to make this clear in the revised paper.

  9. The $W_\infty$ algebra indeed has the automorphism suggested which will be crucial to keep in mind for describing the CFT underlying the critical behaviour of the lattice system. As was mentioned before, this is beyond the scope of our work. We believe that our manuscript possesses some form of completion and we'd prefer not to include an additional speculative discussion that may turn out to be incorrect.

---

## Round 3 · List of Changes

see comments above

---

## Editorial Decision

accepted_in_target_journal